_Article_

# Dissecting the spatiotemporal diversity of adult neural stem cells

Nina Mitic[1,7], Anika Neuschulz [ID][1,2,7], Bastiaan Spanjaard[1], Julia Schneider[3,4,5], Nora Fresmann [ID][1,6], Klara Tereza Novoselc[3,4,5], Taraneh Strunk [ID][1], Lisa Münster[1], Pedro Olivares-Chauvet[1], Jovica Ninkovic [ID][3,4] & Jan Philipp Junker [ID][1,6✉]

## Abstract

Adult stem cells are important for tissue turnover and regeneration. However, in most adult systems it remains elusive how stem cells assume different functional states and support spatially patterned tissue architecture. Here, we dissected the diversity of neural stem cells in the adult zebrafish brain, an organ that is characterized by pronounced zonation and high regenerative capacity. We combined single-cell transcriptomics of dissected brain regions with massively parallel lineage tracing and in vivo RNA metabolic labeling to analyze the regulation of neural stem cells in space and time. We detected a large diversity of neural stem cells, with some subtypes being restricted to a single brain region, while others were found globally across the brain. Global stem cell states are linked to neurogenic differentiation, with different states being involved in proliferative and non-proliferative differentiation. Our work reveals principles of adult stem cell organization and establishes a resource for the functional manipulation of neural stem cell subtypes.

**Keywords** Radial glia; Zebrafish; Single-cell Transcriptomics; Massively Parallel Lineage Tracing; Single-cell RNA Metabolic Labeling
**Subject Categories** Development; Neuroscience; Stem Cells & Regenerative Medicine

## Introduction

Adult stem cells are rare populations that can produce a range of mature cell types to maintain or regenerate tissues. Well-studied examples for adult stem cells include blood and intestine as well as the brain. The vertebrate brain exhibits an extremely high diversity of neuronal cell subtypes and a clear zonation into distinct brain regions (Lake et al, 2016; Zeisel et al, 2018). Neuronal diversity is established in embryonic development, but new neurons can still be generated in the adult brain: While there is limited neurogenic activity in the adult mammalian brain (Gould, 2007; Kaslin et al, 2008), massive neurogenesis has been reported in zebrafish (Adolf et al, 2006; Grandel et al, 2006), which are able to fully regenerate after stab wound injury to the telencephalon (März et al, 2011; Baumgart et al, 2012). Regenerative neurogenesis in zebrafish is driven by radial glia cells, which are the source of new adult neurons and also have tissue maintenance function (Jurisch-Yaksi et al, 2020). Interestingly, adult neurogenesis can proceed via two different paths: direct differentiation without cell division, and proliferative differentiation (Barbosa et al, 2015). Hence, the adult zebrafish brain constitutes an attractive model system for dissecting the functional diversity of adult stem cell states.

While the emergence of neuronal diversity during development has been well characterized by single-cell RNA-seq (Raj et al, 2020), current datasets of the adult zebrafish brain (Cosacak et al, 2019; Yu and He, 2019; Lange et al, 2020) are of lower throughput or restricted to particular brain regions. Therefore, it remains unclear if the radial glia of the adult brain exhibit a diversity of cell subtypes similar to neurons, which might for instance be specific to certain brain regions. Previous work identified diverse spatial signatures of early radial glia cells in the embryonic brain but reduced heterogeneity at larval stages (Raj et al, 2020). Which of these scenarios applies to the adult zebrafish brain remains an open question. Furthermore, radial glia combine functions of both astrocytes and neural stem cells: synapse maintenance, $Ca^{2+}$ buffering, blood-brain barrier maintenance, as well as production of new neurons in the intact and regenerating brain. In addition, radial glia in the telencephalon restrict the lateral ventricle and have the role of ependyma. It is currently not known whether these different functions of radial glia are linked to distinct expression profiles, or whether direct and proliferative differentiation involve the same or different radial glia states. In order to systematically address these issues, we performed a spatiotemporal single-cell RNA-seq analysis of the adult zebrafish brain. We contrasted radial glia in the telencephalon to three other brain regions (diencephalon, mesencephalon, cerebellum) by combining single-cell transcriptomics with massively parallel lineage tracing and in vivo RNA metabolic labeling, in order to decipher the temporal relationships between radial glia subtypes.

[1]Max Delbrück Center for Molecular Medicine in the Helmholtz Association, Berlin Institute for Medical Systems Biology, Berlin, Germany. [2]Humboldt Universität zu Berlin, Institute for Biology, Berlin, Germany. [3]Helmholtz Center Munich – German Research Center for Environmental Health, Institute of Stem Cell Research, Munich, Germany. [4]Biomedical Center Munich (BMC), Department of Cell Biology and Anatomy, Medical Faculty, LMU, Munich, Germany. [5]Graduate School of Systemic Neurosciences, LMU, Munich, Germany. [6]Charité – Universitätsmedizin Berlin, Berlin, Germany. [7]These authors contributed equally: Nina Mitic, Anika Neuschulz. ✉E-mail: janphilipp.junker@mdc-berlin.de

# Results

## Transcriptional diversity in the adult zebrafish brain

For a systematic assessment of radial glia diversity, we performed single-cell RNA-seq of around 100,000 cells from dissociated zebrafish brains (Methods). Computational clustering revealed the expected major cell categories (Fig. 1A–C, Appendix Fig. S1, Dataset EV1): neurons (with high expression of genes related to synaptic activity such as *snap25a*, *syt1a*, and *stx1b*), radial glia (characterized by expression of known markers *fabp7a*, *s100b*, and *cx43* (Than-Trong and Bally-Cuif, 2015)), oligodendroglia (expressing the lineage-specific transcription factor *olig2* as well as *myelin protein zero (mpz)* and *cd59*), and immune cells (expressing *cd74a*, *cd74b* as well as *pfn1*). In addition, several smaller groups of cells were identified that correspond to other niche-specific cell types such as ependymal cells (expressing *epd*, *rbp4*, and *cp*, similar to the ependymal cell type identified in the larval brain (Raj et al, 2020) and in the mouse (Cebrian-Silla et al, 2021)) or cells related to the circulatory system, such as erythrocytes (*hbba1*, *cahz*) and endothelial cells (*lyve1b*, *sox7*). The final group of cells detected in this level of clustering was identified as epithelial cells based on high expression of keratin genes (*krt8*, *krt18*), although the precise location and function of these cells remains unclear. Hereafter we refer to these major categories as "cell types", and we next proceeded to sub-clustering the most abundant groups at a higher resolution in order to fully explore their underlying heterogeneity (hereafter referred to as "cell states" or "subtypes").

The sub-clustering revealed a large diversity of subtypes among neurons (23 identified subtypes) and radial glia (18 subtypes), even in the immune cells (14 subtypes, including microglia, several clusters of macrophages and other circulating immune cells), but not among oligodendroglia (only four clusters identified, three of which correspond to mature or committed oligodendrocyte subtypes and one to oligodendrocyte precursor cells) (Fig. 1D, Datasets EV2, EV3). The clustering resolution was selected so that the majority of clusters could be uniquely distinguished by a set of marker genes, the most characteristic of which is indicated in the assigned name. Furthermore, the biological function or specific name is indicated in case it is known. For example, within the neurons, it was possible to identify several types of granule cells (with pan-granule markers *zic1* and *zic2a*) with additional genes specific to one cluster, such as expression of *tcima* in neuronal sub-cluster 6 or *stxbp6l* in sub-cluster 7. Diversity was also evident at the neurotransmitter level, with several clusters detected for gabaergic (*gad2*, *gad1b*, *slc6a1a*) and glutamatergic (*adcyap1b*, *chga*, *nrgna*) neurons. Importantly, three clusters could be classified as newborn neurons (expressing *tubb5* and elevated *elavl3*), which underscores the fact that neurogenesis can be detected in the adult zebrafish brain even in the baseline healthy state.

For the radial glia (Appendix Fig. S1), sub-clustering revealed one small group of neuroepithelial cells (reduced levels of *fabp7a* and additionally expressing high *vim*, *clu*, and *krt17*) as well as a variety of genuine radial glia subtypes. Some of the identified sub-clusters could be linked to previously described biological states, such as the *prss35*-positive cells (sub-cluster 6), which correspond to Bergmann glia that had been described in the cerebellum (Raj et al, 2020). The *apof*-positive cells are likely a population of radial glia in the optic tectum (expressing *mgfe8a*) (Yu and He, 2019). The

radial glia with *enkur* expression have a similar gene expression profile as a subtype previously found in the telencephalon (Cosacak et al, 2019), which was speculated to be cells in the ependymal lining of the ventricle. Furthermore, some radial glia states showed expression of genes related to proliferation or neurogenic differentiation: markers for S and G2/M phase (*mcm2* and *mki67*); high Notch signaling, which suppresses proliferation (*her4++*); and expression of the synaptic gene *snap25a*.

We additionally carried out GO, KEGG as well as reactome pathway analysis on the marker genes of all radial glia subclusters, with the results confirming our annotation (Methods). *Mcm2* and *mki67* radial glia, for example, show enrichment in multiple cell cycle-related terms and pathways, *her4++* radial glia exhibit an enriched Notch signaling-related term, *stra6+* radial glia show a term confirming their responsiveness to retinoic acid signaling, and *snap25+* radial glia are characterized by multiple neurotransmitter and synapse related GO terms. A full table of the results can be found in Dataset EV4.

For assessing the expression of disease-associated genes in our dataset, we started from a list of genes for which a neuronal disease phenotype was previously reported (Methods). We focused this analysis on the mouse due to its more common use as a disease model. We created a list of zebrafish orthologues and queried the expression of all genes with sufficiently high expression in our radial glia dataset (see Methods for details on gene selection and filtering). From a total of 173 unique mouse genes (Dataset EV5), we ended up with 54 zebrafish genes expressed in our dataset, that are visualized in a dot plot across different radial glia subtypes (Appendix Fig. S2). The resulting highly expressed genes are present across the radial glia subtypes at different frequencies and expression levels. It is notable that a particularly high number of disease-related genes are expressed in radial glia subtypes that are involved in proliferation and differentiation, such as both subtypes of proliferating cells as well as *her4++* radial glia. Many of the detected genes in these subtypes were previously reported to be important for physiology and differentiation of radial glia, such as *her4* or the neural differentiation factor *ascl1a*. Particularly highly expressed in the *mki67+* proliferating cells is the *cfl1* gene, which is a depolymerizing factor for actin and whose mouse ortholog was shown as crucial for migration of radial glia and establishment of cortical layers (Bellenchi et al, 2007). Structural genes of the cytoskeleton appear as important in other cell types as well, such as the broadly expressed *actb1* and *actb2* as well as the canonical radial glia marker *gfap* (in our dataset only highly expressed in *enkur+* radial glia) and *vim* (neuroepithelial cells). Among the remaining genes, it is interesting to highlight the members of the inhibitor of DNA binding family that inhibit binding of transcription factors (*id1*, *id2a*, and *id3*), whose mouse orthologs are important for anchorage of stem cells in their specific niche (Niola et al, 2012). Interestingly, the different variants are expressed in different radial glia subtypes, which is in line with the mouse data where, for example, *id2* has been specifically identified as important for development of dopaminergic neurons for olfactory neurogenesis (Havrda et al, 2008). Another specifically expressed gene is *nrg1* (neuregulin 1), a marker gene for *nrg1+* radial glia in our dataset, which has been shown to have a crucial role in conversion of radial glia to astrocytes in mouse (Schmid et al, 2003).

In summary, while the existence of a large diversity of neuronal subtypes had already been firmly established, we observed a similar degree of heterogeneity among radial glia in the adult zebrafish

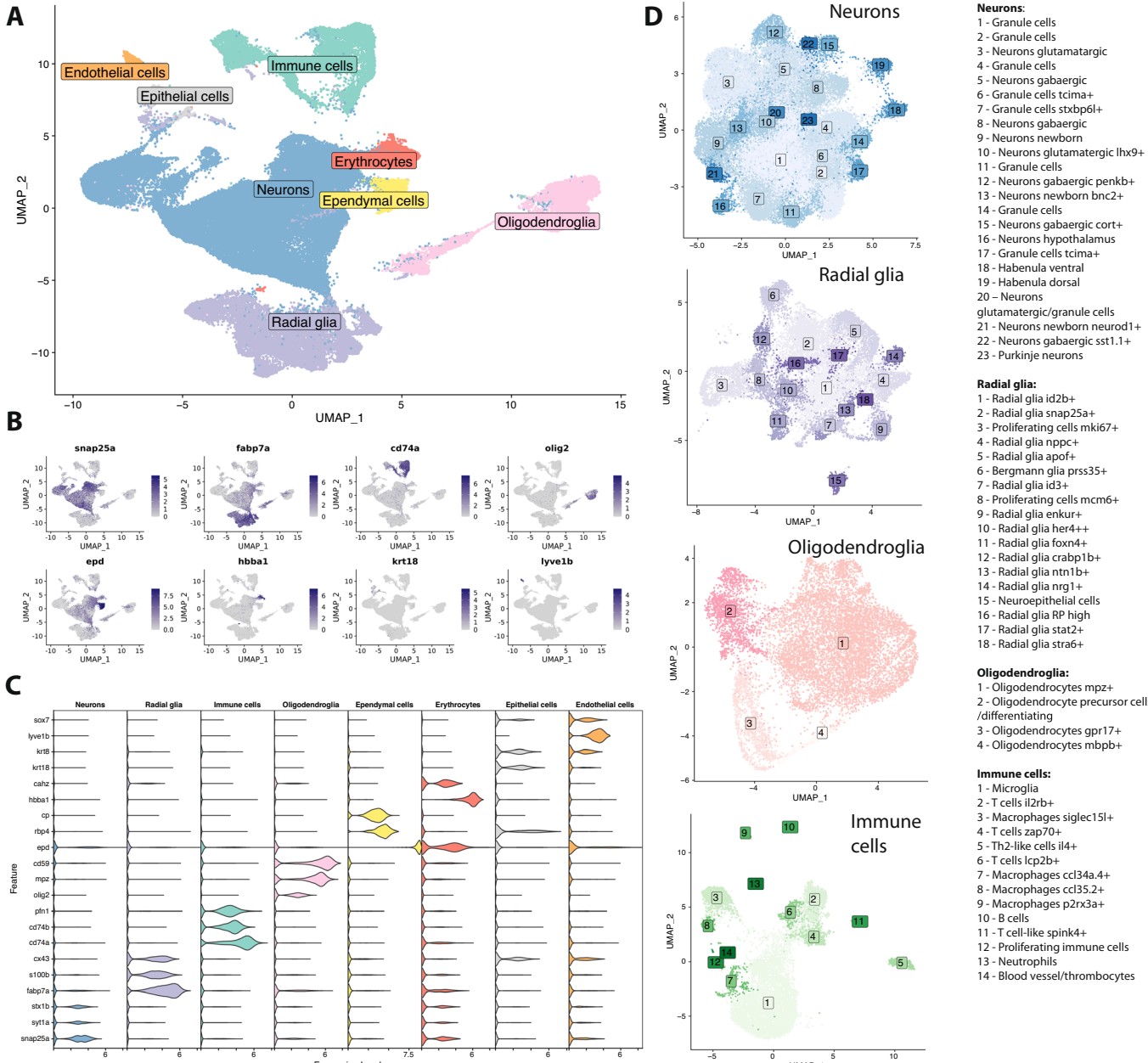

**Figure 1. Transcriptional diversity in the adult zebrafish brain.**

(A) UMAP representation of the scRNA-seq dataset of the adult zebrafish brain, with color indicating major cell types. The dataset consists of 107,698 single cells sampled from whole brains, dissected specific brain regions and whole brain cells enriched for *gfap*-positive cells. 20 biological replicates were prepared and one animal per library was used in each replicate (with the exception of FACS sorted and methanol fixed samples, where cells from multiple animals were pooled). (Also see Appendix Fig. S1 and Dataset EV1 for library composition. Sample IDs b1 to b20 were used in this figure). (B) UMAP plots highlighting expression of individual representative marker genes for each of the major cell types. Expression levels based on scaled data. (C) Violin plots of additional marker genes for the major cell types. Expression levels based on scaled data. (D) Sub-clustering of the four most abundant major cell types (neurons—64,582 cells, radial glia—15,829 cells, oligodendroglia—10,949 cells and immune cells—9345 cells) reveals a variety of subtypes and states. The identified clusters are depicted in the UMAP representation of each subclustered cell type, with the assigned name in the text box. See Dataset EV2 for a list of marker genes for all subtypes.

brain. However, the function of these different radial glia states remains largely unexplored. To dissect the organizing principles underlying radial glia diversity, we therefore decided to investigate their spatial partitioning, developmental origin, and involvement in neurogenic differentiation trajectories.

## Spatial distribution of radial glia across different brain regions

We first hypothesized that radial glia diversity might be related to regional specialization and adaptation to different anatomical

regions, as recently described for astrocytes in the mammalian system (Ohlig et al, 2021). To investigate this idea, we performed dissections into four major anatomical territories, telencephalon, diencephalon, mesencephalon and cerebellum, followed by single-cell RNA-seq. In addition, we complemented these experiments with scRNA-seq of undissected brains and FACS enrichment of radial glia cells using the *gfap:GFP* reporter line. While there were pronounced differences in relative abundance, all major cell classes could be detected across the four anatomical regions (Appendix Fig. S3A,B). As a control, we confirmed that each cell class could be found in dissected samples as well as in whole brain dissociations (Appendix Fig. S3A,B), and we used the subset of whole brain samples to validate that the detected radial glia subtypes were not dominated by batch effects (Appendix Fig. S3C). We then proceeded to determine the anatomical distribution of the detected subtypes in order to distinguish between global cell states (found across multiple brain regions) and regional cell states (enriched in one brain region). While neuronal subtypes were mostly highly specific to individual brain regions, along with a few globally distributed subtypes, the oligodendroglia and immune cells generally did not display a pronounced anatomical enrichment (Fig. 2A). Interestingly, the radial glia fell into an intermediate category, with some states being largely restricted to one anatomical region ("regional"), while others were spread globally across the brain (Fig. 2A). We therefore assigned each of the 18 radial glia states as being either "regional" or "global" based on whether the maximum fraction in one anatomical region was above or below 70% of the total (see Methods). We set the cutoff to 70% to account for the limited accuracy of dissection, as evidenced by our data: For example, the known cerebellar population of Bergmann glia (cluster 6) is accurately found in the rhombencephalon but only at ~80%, suggesting that a certain degree of cross-region contamination does occur and the cutoff needs to be permissive. We found that, among the four brain regions, dissection of the diencephalon was the least accurate.

Of note, all of the global radial glia states were found in all four anatomical regions—in contrast to global neurons, which can also be restricted to two or three regions (e.g. sub-cluster 7). Furthermore, our designation of "global" radial glia refers exclusively to the presence of these clusters across all four brain regions. This does not preclude that these cell states may display a spatially restricted localization within an anatomical region, and this does not necessarily imply a difference in lineage origin.

Two of the nine global radial glia states were characterized by a clear proliferation signature (Fig. 2B), suggesting that these cell states are linked to proliferative differentiation. By contrast, another global radial glia state (*her4++* radial glia) exhibited high expression of genes involved in Notch signaling (Fig. 2B; Appendix Fig. S3D), which is known to suppress proliferation and differentiation of radial glia (Chapouton et al, 2010). These observations raise the possibility that at least some of the global radial glia states could contribute to neurogenesis.

To validate our approach for identification of region-specific radial glia subtypes based on dissection of brain regions and scRNA-seq, we performed fluorescent in situ hybridization (RNAscope) for selected marker genes. We found that the microscopy results are in full agreement with our sequencing data, thereby validating the approach (Fig. 2C,D; Appendix Fig. S4). Specifically, we found that, as expected, expression of *apof* was

restricted to the mesencephalon, *prss35* was only detected in the rhombencephalon, AL954697.1 was specific to the diencephalon, and *crabp1b* was only detected in the rhombencephalon. Furthermore, we also used RNAscope to validate the existence of *snap25+* radial glia. Since the marker gene *snap25* is also expressed in neurons, we wanted to rule out the possibility that *snap25+* radial glia might be an artifact caused by ambient RNA or doublets containing a neuron and a radial glia cell. We observed co-expression of the marker genes *snap25* and *ggctb* in a subset of radial glia, which validated the detected population of *snap25+* radial glia (Appendix Fig. S4).

## Developmental emergence of radial glia diversity

We next hypothesized that the observed diversity of radial glia states might be related to early developmental lineage separation. To determine to which degree radial glia diversity is already present at embryonic and larval stages, we integrated our data with a previously published dataset (Raj et al, 2020) covering developmental neurogenesis (Fig. 3A). To ensure a general overlap of cell states, we omitted the very early embryonic progenitors that had a different gene expression profile, as well as retinal progenitors, and selected for our analysis only radial glia and neuronal progenitors of the brain between 20 h and 15 days of larval development. These cells were integrated with the radial glia of the adult brain using canonical correlation analysis (Stuart et al, 2019), and the relative contribution of each time point to shared clusters was used as an indication of similarity. We found that the developmental dataset integrated well into our adult data, with contributions of embryonic or larval cells to all adult clusters, albeit at different proportions (Fig. 3A). The most notable difference was that proliferating radial glia were strongly enriched at embryonic and larval stages (Fig. 3A). Furthermore, we observed that the abundance of some additional radial glia states (e.g. *her4++* and *id2b+* radial glia) decreased, while others (e.g. *snap25a+* and *enkur+* radial glia) increased in adult fish compared to embryonic/larval stages. In summary, we found that radial glia diversity is already established at embryonic and early larval stages.

We further asked whether global radial glia might have a different developmental origin from regional radial glia. To add information about the developmental origin of adult brain cells, we applied a method for massively parallel lineage tracing based on CRISPR-Cas9 technology (Spanjaard et al, 2018; Raj et al, 2018; Alemany et al, 2018). By injecting Cas9 and single-guide RNAs (sgRNAs) against the 3' UTRs of housekeeping genes, we recorded lineage relationships in early development (from fertilization until gastrulation (Spanjaard et al, 2018)) by creating 'genetic scars' that serve as lineage barcodes (Fig. 3B, Methods). As targets for lineage barcodes, we selected genes that are highly and ubiquitously expressed across cell types and where creation of genetic scars did not cause an observable phenotype in the fish (*actb1, actb2, cfl1, cirbpb, rpl39, ube2e1*). To identify groups of mitotically related cells, we harnessed the fact that each cell can have at most two distinct scar alleles per target gene. On a per-target basis, we first identified pairs of sequences that are consistently observed together, yielding a list of allowed sequence combinations. We then filtered for cells that could be assigned such a sequence combination on at least two target genes (Methods). The set of filtered cells were hierarchically clustered based on information from all targets,

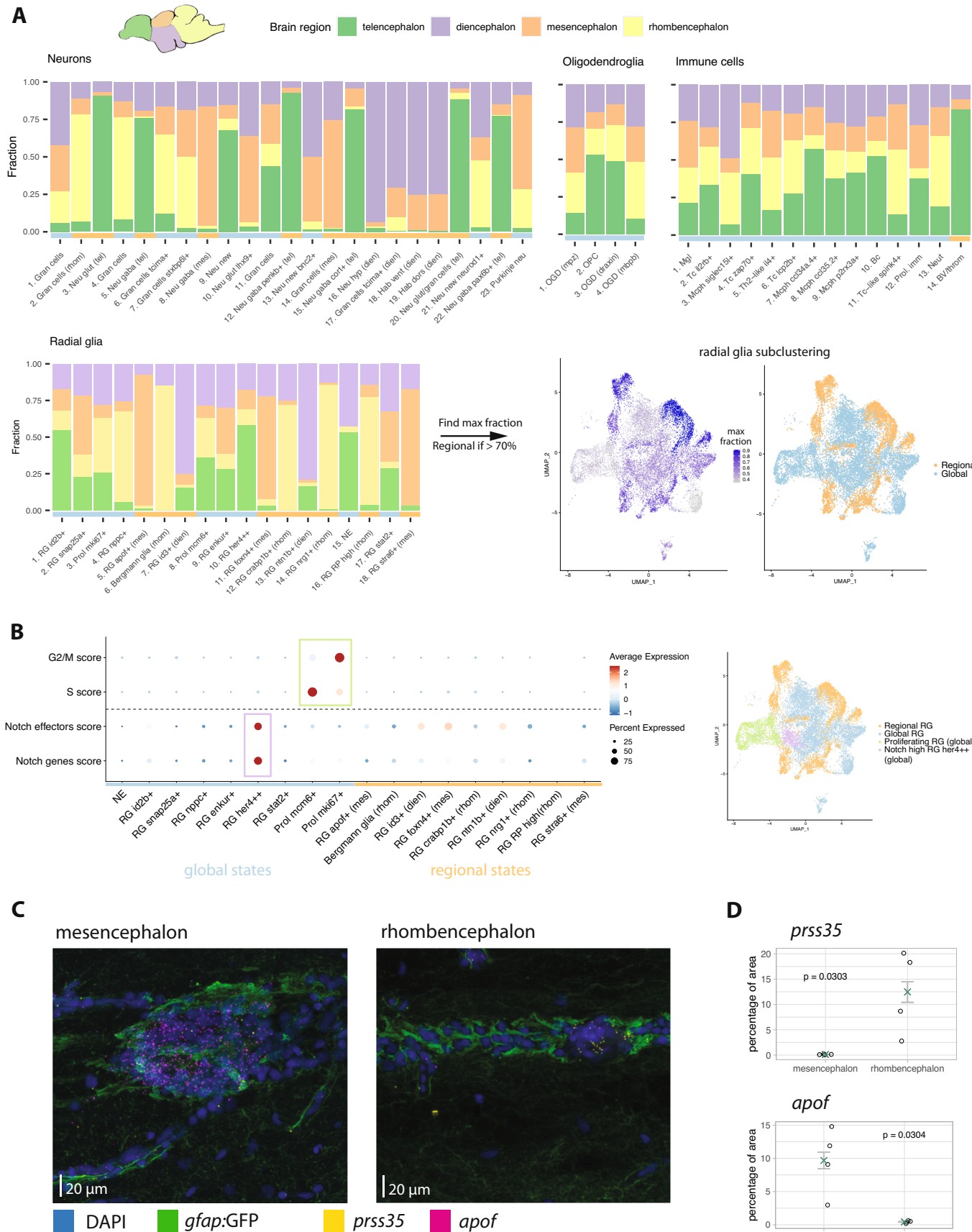

**Figure 2.  Spatial distribution of radial glia across different brain regions.**

(A) Bar plot depicting the spatial origin of each of the subtypes of neurons, oligodendroglia, immune cells and radial glia identified after sub-clustering (cell states shown on the x-axis, with cluster number corresponding to Fig. 1D). The fraction on the y-axis shows the contribution of each dissected brain region to a given cell subtype, after excluding the cells from undissected whole brain libraries and from sorted samples. Cell states that were composed of 70% or more of a single region were classified as regional, while the remaining cell states are considered global (category depicted in the color bar below the bar plot: orange = regional, blue = global). The UMAP representations of subclustered radial glia show the maximum fraction of most abundant brain region for each cluster (left), as well as the binary classification into regional and global states (right). Only dissected samples were used for this analysis (sample IDs b1-b7, b9, b10 and b18, see Dataset EV1). Cell subtype and region names shown as abbreviations: gran cells granule cells, neu neurons, glut glutamatergic, gaba gabaergic, new new newborn neurons, hyp hypothalamus, hab habenula, OGD oligodendrocyte, OPC oligodendrocyte precursor cells, mgl microglia, Tc T cells, Bc B cells, Mcph macrophages, prol proliferating, bv blood vessel, throm thrombocyte, RG radial glia, NE neuroepithelial. (B) Gene module scores for proliferation signal (separated by cell cycle phase) and Notch activity (pathway genes and downstream effectors—see Appendix Fig. S2C for individual representation of genes that contribute to the Notch scores). The cells are grouped by their category as global or regional, as indicted by the color bar below the plot. The groups of cells with high scores (green rectangle for proliferation, purple rectangle for Notch scores) are additionally highlighted in the UMAP representation of subclustered radial glia on the right. (C) RNAscope in situ hybridization. Medial sections of mesencephalon (left) and rhombencephalon (left). Nuclei are stained in blue (DAPI), radial glia in green (gfap:GFP). Expression of prss35 (yellow) and apof (magenta) are shown using RNAscope probes. Four animals were analyzed, with one representative section per brain region. All sections were imaged under identical conditions for quantification.
(D) Quantification of RNAscope signal. Dot plot depicting the fraction of gfap:GFP positive area covered with apof and prss35 signal. The green cross indicates the mean ± SEM. Each circle represents one single animal (one representative section was analyzed per animal). The significance was tested using Mann-Whitney test and the p-value is indicated on the plot.

grouping closely related cells together into lineage clones (Fig. 3C; Appendix Fig. S5A–C).

Focusing on neurons and radial glia, we observed that most lineage clones displayed a clear spatial structure, with clones being dominated by cells from one or at most two brain regions (Fig. 3C, see close-up of selected clones at the bottom). This likely reflects the separation of cells that give rise to different brain regions in the gastrulation-stage fate map (Kimmel et al, 1990). Of note, our lineage analysis was performed with undissected whole brains and therefore does not contain direct spatial information: While we can infer the spatial location of cells in regional states, we do not have information about the spatial location of cells in global states. Our interpretation of the spatial structure of clones is hence purely based on regional states.

Furthermore, we found that, across all clones, radial glia in global states were intermixed with radial glia in regional states (Fig. 3C; Appendix Fig. S5C). This finding shows that there is no early developmental lineage split between global and regional radial glia. This observation would be compatible with a scenario where, within each brain region, radial glia can switch from a regional to a global state, or vice versa.

## Direct and proliferative differentiation follow separate trajectories and involve distinct radial glia states

To follow up on these observations, we sought to investigate the involvement of specific radial glia states in neurogenic differentiation. Specifically, based on the detected gene expression signatures related to proliferation and Notch signaling in global radial glia (Fig. 2B), we hypothesized that at least some of the global radial glia states might be linked to differentiation, while the regional radial glia states might rather be associated with tissue maintenance functions. Furthermore, we speculated that direct differentiation and proliferative differentiation might involve different radial glia states. To gain insight into these questions, we focused on the telencephalon, the experimentally most accessible and therefore best studied of the four brain regions. In a UMAP representation of the scRNA-seq data we found two separate connections between radial glia and neurons, which involved different radial glia states

(Fig. 4A). RNA velocity based on detected intron/exon ratios (La Manno et al, 2018; Bergen et al, 2020) qualitatively supported the interpretation of the connecting zones as two separate trajectories from radial glia to neurons (Fig. 4A). One of these trajectories involved radial glia proliferation, while the other did not. While this is in agreement with the previous observation of two types of neurogenesis in the telencephalon (Barbosa et al, 2015) (direct differentiation versus proliferative differentiation), our analysis provides evidence that the two modes of neurogenesis proceed via separate differentiation trajectories.

We noticed that the 70% threshold used in Fig. 2 for assigning radial glia to global and regional states led to a labeling of almost all radial glia in the telencephalon as global. However, a more nuanced analysis revealed that the radial glia involved in both types of neurogenic transitions belonged to less region-specific subtypes, whereas the radial glia that were not part of these transitions displayed more pronounced enrichment in the telencephalon (Fig. 4A), in line with the idea that global radial glia states are involved in neurogenic differentiation. The subtypes of snap25a+ and her4++ radial glia were of particular interest, since they appeared to be involved in direct and proliferative differentiation, respectively.

We quantified this observation in two ways: First, we determined for each cell its most likely transition to another cell based on RNA velocity transition probabilities (Methods), which confirmed that her4++ radial glia can transition to newborn neurons via proliferating radial glia, while snap25a+ radial glia can transition directly to neurons but not to proliferating radial glia (Fig. 4B). Next, we expanded this approach beyond a one-step transition towards trajectory reconstruction by using CellRank (Lange et al, 2022). CellRank combines identification of stable sets of cells in gene expression states with diffusion pseudotime and RNA velocity to calculate cell state transition probabilities (Methods). This approach confirmed possible differentiation paths from her4++ radial glia to proliferating radial glia and to newborn neurons, and from snap25a+ radial glia to neurons without proliferation (Fig. 4C; Appendix Fig. S6A).

A crucial difference between the two analyses in Fig. 4B,C is that CellRank attempts reconstruction of full trajectories. In agreement with our interpretation, the CellRank analysis identifies a large fraction of

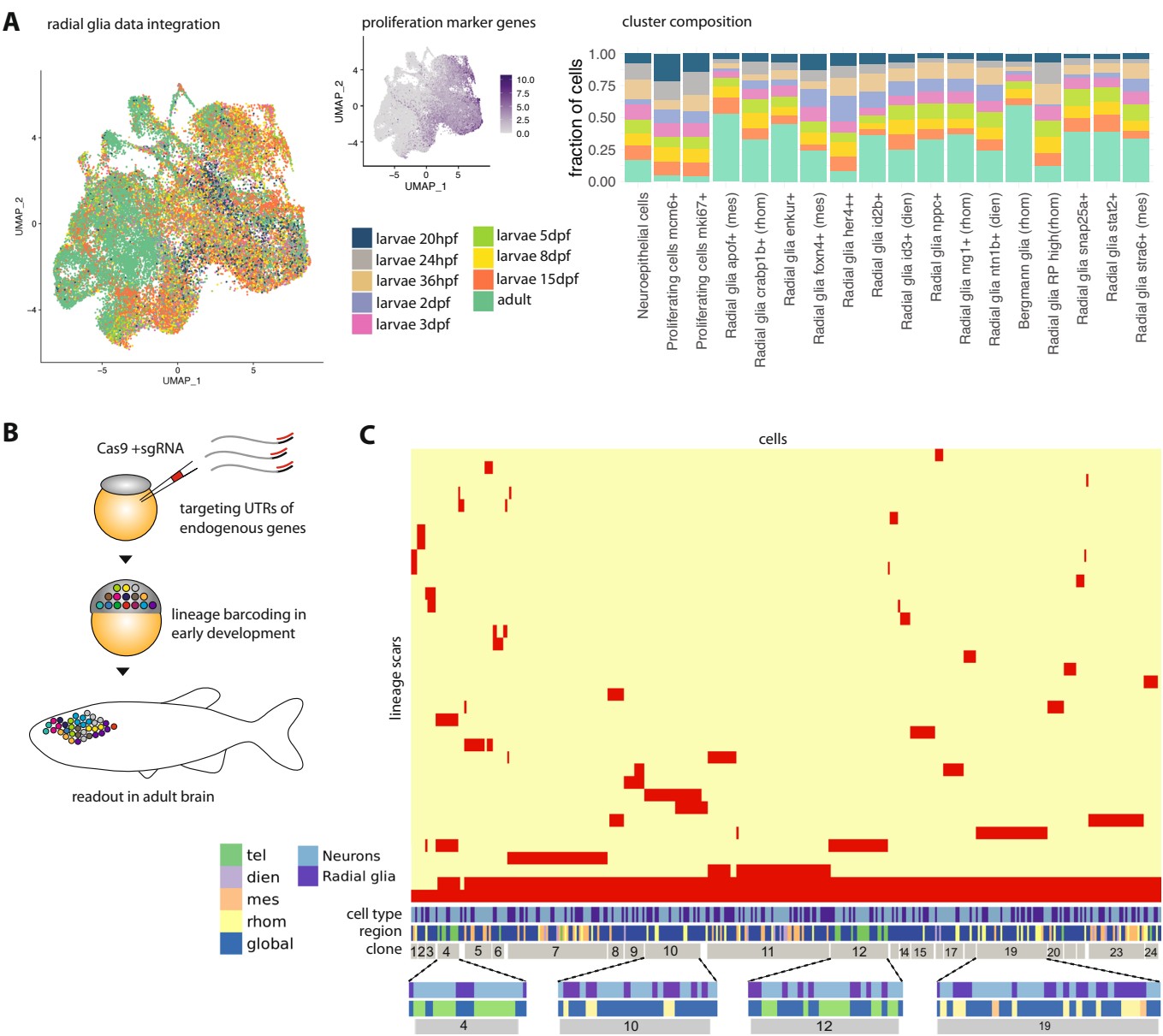

**Figure 3. Developmental emergence of radial glia diversity.**

(A) Data integration of adult radial glia with developmental radial glia and neural progenitors from a published dataset (Raj et al, 2020). The datasets were integrated with canonical correlation analysis, and the resulting joint embedding is shown in the UMAP on the left. To assess the time point at which specific cell states may have emerged, for each adult radial glia subtype the co-clustering larval stages were quantified (see Methods). Adult radial glia from sample IDs b1 to b20 were used in this analysis (see also Dataset EV1). (B) Sketch of the experimental approach for lineage tracing. Lineage barcodes were introduced during early development by injection of Cas9 and sgRNAs targeting untranslated regions of highly expressed genes. The fish were grown to adulthood and the lineage barcodes were read out along with the transcriptome from single cells of the adult brain. (C) After sequencing, lineage barcodes were filtered for quality and then clustered by similarity to reveal lineage relationship between the cell (sub)types (see Methods). The depicted heatmap shows hierarchical clustering of radial glia and neurons based on identified lineage scars for 790 cells from a single brain. Clones were identified based on shared scar barcodes and are indicated in gray at the bottom. We found that clones typically contain a mixture of global and regional states, with the latter originating from a single brain region. Very small clones and cells without detected scars were not annotated as clones. Sample ID lintrace_rep1 is shown here (see Dataset EV1).

transitions from *her4++* radial glia to neurons, thereby confirming neuronal differentiation as the final stage of the trajectory (Fig. 4C). By contrast, the dominant non-self transition of *her4++* radial glia in the one-step analysis is towards proliferating cells (Fig. 4B), which validates the role of *her4++* radial glia in proliferative differentiation. Of note, the two analyses in Fig. 4B,C also reveal additional transitions (besides

*snap25+* radial glia to neurons and *her4++* radial glia to proliferating cells to neurons). This is to be expected for two reasons: First, the transitions are not directly measured but inferred based on transcriptomic similarity and RNA velocity, so a certain level of background signal is unavoidable. Second, and more importantly, we here analyze steady-state conditions in the unperturbed adult brain with only a limited amount of

## A  telencephalon

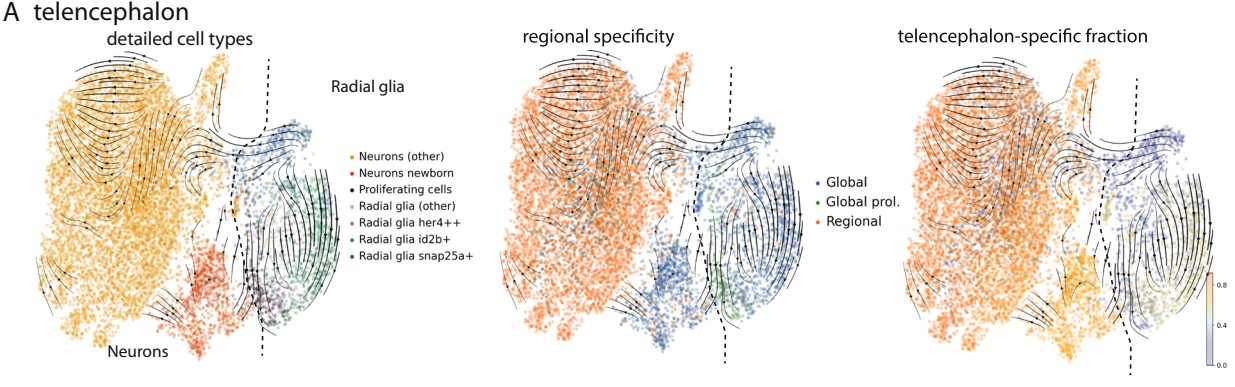

## B  transition probability from each cluster

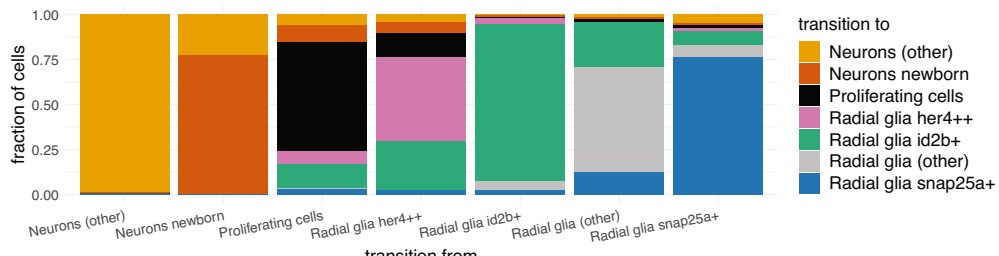

## C  CellRank absorption probabilities for each cluster

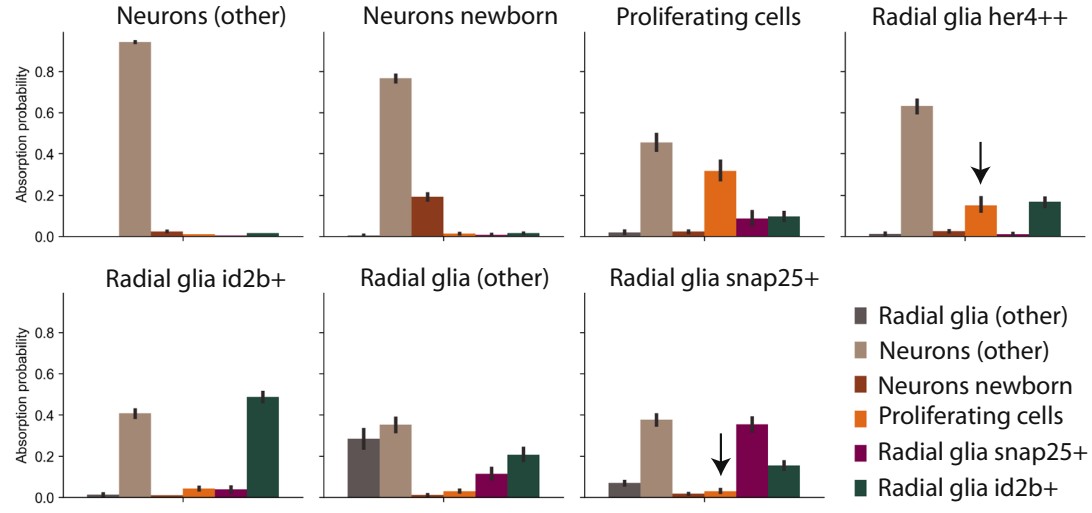

## D

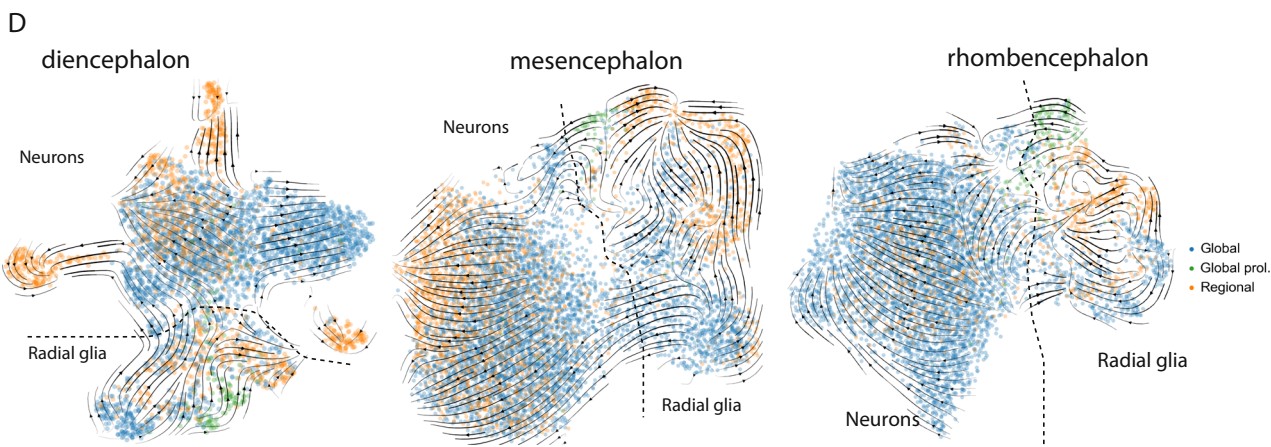

**Figure 4.  Direct and proliferative differentiation follow separate trajectories and involve distinct radial glia states.**

(A) UMAP embeddings of combined telencephalon dataset in the healthy brain (9406 cells, taken from sample IDs b1-b7, see Dataset EV1) subsetted to radial glia and neurons, with velocity embedding streams derived from the stochastic model of splicing based velocities in scVelo. Left: Subtypes of relevance for neurogenesis are highlighted among radial glia and neurons. Middle: regional specificity classification of cells based on the 70% cutoff as described in Fig. 2A, with two main cell types (radial glia and neurons) indicated by a dashed line. Right: regional specificity score for telencephalon, based on the calculation from the whole brain dataset.
(B) Transition probabilities (based on RNA velocity) between highlighted cell states. For each cell within the cell state, the maximum transition probability and the corresponding cell state to which that cell is most likely to transition were determined. The bar charts represent the fractions of transition probabilities for all cells of a given cell state towards all the other cell states, including self-transitions. (C) Prediction of absorption probability for each of the highlighted cell state groups based on the CellRank algorithm. The absorption probabilities describe the predicted transitions from one cluster to all other clusters ("terminal states"). We detect possible differentiation paths from $her4++$ radial glia to proliferating radial glia and neurons without involvement of $snap25+$ radial glia, and from $snap25+$ radial glia to neurons without proliferation (see black arrows). Error bars show 95% confidence interval on mean absorption probability over all cells ($n = 9406$). (D) UMAP embedding with indicated main cell type and regional specificity classification for other brain regions: diencephalon (3970 cells, 3 biological replicates), mesencephalon (7590 cells, 2 biological replicates) and rhombencephalon (5917 cells, 3 biological replicates). The main cell types (radial glia and neurons) are indicated by a dashed line. Sample ID b18 was used in this analysis (see also Dataset EV1).

ongoing neurogenesis. It is therefore not surprising that, in addition to neurogenic differentiation, we also detect other transitions, including from global to regional radial glia states.

We also observed a separation into two neurogenic trajectories in the mesencephalon and cerebellum (Fig. 4D, Appendix Fig. S6B), indicating that the pathways for direct and proliferative differentiation coexist across the brain and that the role of $her4++$ and $snap25a+$ radial glia may be conserved across the brain. In summary, this analysis suggests that global radial glia states tend to be linked to neurogenic differentiation, while regional radial glia states might have other functions. More specifically, our data indicates that $snap25a+$ radial glia might be involved in direct differentiation, while $her4++$ radial glia were inferred to be directly upstream of proliferative radial glia.

## In vivo single cell RNA metabolic labeling validates the observed trajectories

The UMAP layout in Fig. 4A showed two different connections between radial glia and neurons, which involve different radial glia states and which we interpreted, in agreement with the literature (Barbosa et al, 2015), as two separate differentiation trajectories corresponding to direct and proliferative differentiation, respectively. However, RNA velocity inference based on intron/exon ratio has conceptual limitations and is prone to creating artefacts due to e.g. simplifying model assumptions and low intron detection in scRNA-seq (Bergen et al, 2021; Gorin et al, 2022; Marot-Lassauzaie et al, 2022). We therefore set out to validate the directionality of the two trajectories from radial glia to neurons in another way. However, lineage tracing based on genetic markers, using for instance Cre/lox technology, is difficult in this system: Suitable marker genes are scarce, since expression differences between radial glia subtypes are often of a gradual nature. For instance, there are no good markers for the $her4++$ radial glia, which are characterized mostly by high Notch signaling and by absence of regional markers (Appendix Fig. S1). We therefore decided to validate the state transitions in the telencephalon with a complementary approach that does not require marker genes, by using time-resolved single-cell RNA-seq via metabolic RNA labeling (Fig. 5A). We reasoned that, due to the uncertainties associated with intron/exon based RNA velocity, it would be more robust to base velocity information on experimental measurement of the RNA molecules that are transcribed in a specific time

window. Using the scSLAM-seq method (Erhard et al, 2019; Qiu et al, 2022; Cao et al, 2020; Holler et al, 2021), we labeled newly made transcripts for 6 h, and then proceeded to scRNA-seq analysis (Fig. 5A). The experimental approach included intraventricular injection of 4-thiouridine (4sU), methanol fixation of cells, and a thiol-modification step that converts 4sU labels into T-to-C conversions on the level of the cDNA (Methods). We found that this approach specifically increased the fraction of T-to-C conversions compared to control samples (Fig. 5B). This approach has at least two advantages compared to analysis of intron/exon ratios: (1) The computation is based on all detected genes, not only on those with introns that are detectable by scRNA-seq. (2) The time window of labeling is well defined and is the same for all genes, in contrast to gene-dependent splicing rates. However, other potential computational issues related to model biophysics, fitting, normalization and scaling remain to be addressed (Bergen et al, 2021; Gorin et al, 2022; Marot-Lassauzaie et al, 2022).

In addition to scSLAM-seq in unperturbed conditions, we also inhibited Notch signaling by a 48-hour incubation in a water bath with the γ-secretase inhibitor DAPT (Chapouton et al, 2010) (Appendix Fig. S7A, Methods) with the goal to modify the ratio of direct versus proliferative differentiation. We labeled mRNA from 42 h until 48 h after the start of the treatment, followed by preparation of a single cell suspension for scSLAM-seq (Methods). Using a recently established computational method for calculating RNA velocity based on the combination of labeled/unlabeled and intron/exon reads (Qiu et al, 2022) (Methods), we confirmed the directionality of the two trajectories (proliferative versus non-proliferative differentiation) (Fig. 5C). On a technical level, we observed slight changes in transcriptome profiles in scSLAM-seq compared to the conventional scRNA-seq data, which might be related to the methanol fixation step in scSLAM-seq, and which interfered with computational integration of the two data types. We therefore decided to analyze the scSLAM-seq data separately, and we annotated cell states based on marker genes (Appendix Fig. S7B). When comparing the two conditions (Notch inhibition versus control) in separate UMAP embeddings, we did not detect major changes in cell state abundance and noticed only slight variations in the fraction of $snap25+$ radial glia involved in direct differentiation (Fig. 5D). While we can speculate that the observed increase in direct differentiation after Notch inhibition might be a compensatory effect following the induction of proliferative differentiation, more experiments would be needed to substantiate this point.

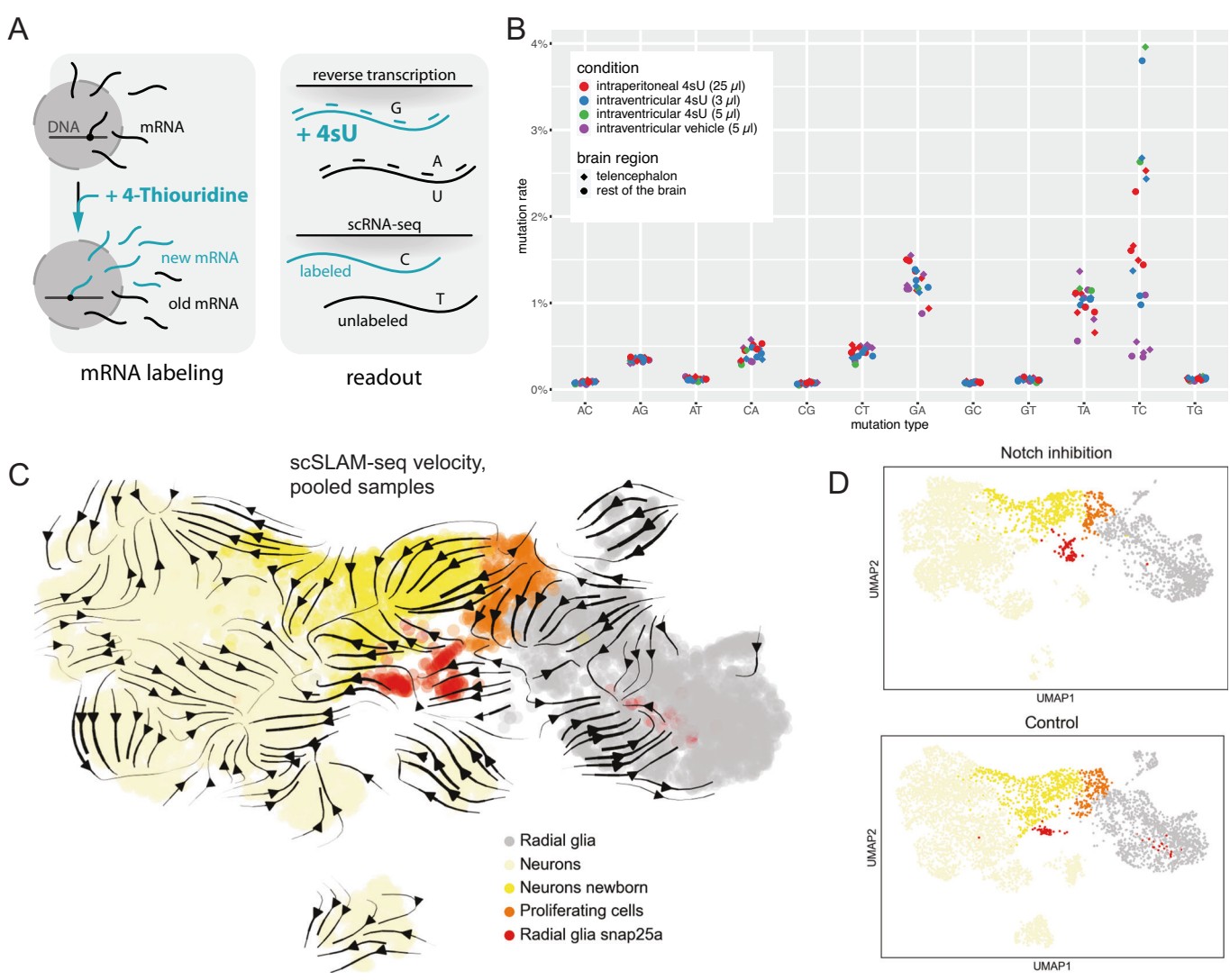

Figure 5. In vivo single cell RNA metabolic labeling validates the observed trajectories.

(A) Schematic depiction of the scSLAM-seq method. The uridine analogue 4-Thiouridine (4sU) is delivered to the cell and incorporated into newly transcribed RNA molecules. During the reverse transcription reaction, the incorporated 4sU leads to a mismatch in base pairing, which is read out in the final library as a mutation of thymidine into cytosine (T-C mutation). (B) Observed mutation rates for all nucleotides in different conditions of 4sU delivery to the brain, based on bulk sequencing readout. The brain samples were split into telencephalon and the rest of the brain to assess the local diffusion of 4sU after delivery. (C) Combined UMAP embedding of two scSLAM-seq datasets: chemical Notch inhibition (incubation with DAPT) for 48 h and the control (incubation with DMSO). For both control and treatment sample, cell state identities were assigned based on marker gene expression (see Appendix Fig. S5B). Velocity embedding streams were calculated based on combined splicing-based and labeling-based velocities (see Methods). Sample IDs b22 and b23 are shown here (see Dataset EV1). (D) Separate UMAP embeddings for Notch inhibition (6353 cells, 1 biological replicate) and control (4630 cells, 1 biological replicate) datasets. Notch inhibition is known to induce proliferative differentiation. However, this did not manifest in clear changes in cell state abundance.

## Discussion

Here, we performed a spatiotemporal dissection of radial glia states in the adult zebrafish brain by combining single-cell RNA-seq with dissection of different brain regions, comparison to embryonic and larval stages, massively parallel lineage tracing, and single-cell RNA metabolic labeling. Our analysis of RNA velocity based on scSLAM-seq is the first demonstration of single-cell RNA metabolic labeling in a live adult vertebrate, which here allowed us to validate cell state transitions and differentiation trajectories. We anticipate that in vivo RNA metabolic labeling will emerge as a powerful approach for determining the

directionality of cell state changes. In summary, the combination of experimental and computational approaches presented here can serve as a blueprint for functional dissection of stem cell state diversity in complex tissues.

We found that some radial glia states exist globally across the brain, while others are restricted to specific regions. The diversity of radial glia states is already established at embryonic stages, but CRISPR lineage tracing revealed that global radial glia do not have a separate developmental source. By contrast, we found that clones are typically restricted to specific brain regions. Taken together, these findings suggest that, within each brain region, transitions

between regional and global states are possible. Using RNA velocity (based on intron/exon ratio as well as RNA metabolic labeling) we could show that neurons are only derived from global radial glia states. Regional radial glia states might predominantly have other functions, for instance related to tissue maintenance. We observed that direct and proliferative differentiation proceed along separate trajectories and involve different radial glia states: *snap25+* radial glia are involved in direct differentiation, while *her4++* radial glia give rise to proliferating radial glia. We speculate that *her4++* radial glia are poised for proliferative differentiation and that high Notch signaling, which is known to inhibit proliferation of radial glia, is required to keep them in a non-proliferative state. Of note, we found that *snap25+* radial glia are depleted at embryonic stages compared to adult fish, while *her4++* radial glia and proliferating radial glia are enriched in embryos compared to adults. This observation would be in line with a requirement to prioritize proliferative differentiation in early development in order to expand the number of neurons while maintaining the pool of radial glia. In summary, our combined analysis supports a scenario where radial glia in different brain regions are derived from different clones and have regional identities, but can transition into global states that are linked to neurogenic differentiation.

A limitation of our study is that the dissection approach provides only coarse spatial information, and it is likely that spatial analysis with higher resolution would reveal further biological functions of the radial glia states. Furthermore, our lineage analysis was limited to recording at early development, and it would be interesting to extend the time window of lineage recording in order to identify lineage splits that may occur at later stages. Our study revealed cell state plasticity between at least some radial glia states, and with the detailed transcriptomic analysis presented here, we now hold the key to understanding the underlying regulatory mechanisms. We expect that this knowledge can be used in the future to manipulate neurogenic differentiation by independently inducing proliferative and non-proliferative differentiation.

# Methods

## Ethics statement

This study complied with all relevant ethical regulations. Zebrafish were bred, raised and maintained in accordance with the FELASA guidelines (Aleström et al, 2020), the guidelines of the Max Delbrück Center for Molecular Medicine and the Helmholtz Center Munich, and the local authorities for animal protection (Landesamt für Gesundheit und Soziales, Berlin, and Regierung von Oberbayern, Munich, Germany) for the use of laboratory animals, based on the current version of German law on the protection of animals and EU directive 2010/63/EU on the protection of animals used for scientific purposes. In addition, housing and breeding standards followed the international 'Principles of Laboratory Animal Care' (NIH publication no. 86-23, revised 1985). Data collection and analysis were not performed blind due to the conditions of the experiments.

## Zebrafish lines

The zebrafish strains used in the project were the wild-type AB strain as well as the transgenic lines *Tg[ubi:zebrabow-M]* [a131Tg]

(Pan et al, 2013), Tg[*gfap:GFP*] [mi2001] (Bernardos and Raymond, 2006) and Tg[*ubi:cas9,U6:sgRNA(dTomato), cmlc2:EGFP*] (this publication). Adult zebrafish of random sex that were between three months and one year of age were used for the experiments.

## Preparation of single-cell suspensions from adult brain

Adult zebrafish were sacrificed by immersion in ice-cold water (0–4 °C) for 20 min, in accordance with ethics regulations. The brain was extracted and placed into a dish with cold Hank's Buffered Salt Solution (HBSS). Depending on the type of library, the dissociation was carried out using the whole brain, or the brain was previously manually microdissected to separate the region of interest: telencephalon, diencephalon, mesencephalon or rhombencephalon. The dissection of the diencephalon was less precise than for the other brain regions, since it has three borders to the neighboring brain regions (see cartoon in Fig. 2A), and none of these borders is very clear. By contrast, the borders between telencephalon, mesencephalon and rhombencephalon are much clearer. Two dissociation methods were used to obtain the single-cell suspension: for the standard scRNA-seq samples dissociation was performed with trypsin, while for the scSLAM-seq samples a papain-based method was used. For dissociation with trypsin, the brain tissue was placed in 750 µl of HBSS-Glucose solution in a 1.5 ml tube (to obtain a 500 ml stock of the solution that was stored at −20 °C, we combined 50 ml 10× HBSS (Life Tech), 9 ml 45% D-Glucose (Sigma-Aldrich) and 7.5 ml 1 M HEPES (Gibco), and adjusted pH to 7.5). To initiate dissociation 15 µl of 0.05% Trypsin-EDTA (Gibco) was added and gently mixed with a BSA-coated glass pipette. The suspension was incubated for 30 min at 37° with intermittent pipette mixing. The reaction was stopped by adding 750 µl of a BSA-EBSS-HEPES solution (for a 500 ml stock stored at −20 °C, we dissolved 20 g of BSA (Sigma-Aldrich A4503) in a solution of 490 ml 1× EBSS (Gibco) and 10 ml 1 M HEPES (Gibco) and adjusted pH to 7.5). The suspension was mixed and filtered through a 100 µm filter to remove larger tissue pieces and then centrifuged for 5 min at $300 \times g$ at 4 °C. The cells were washed once with 1 ml cold HBSS and centrifuged again. The final pellet was resuspended in 100–200 µl cold HBSS with 0.05% BSA and filtered through a 35 µm diameter cell strainer, after which the cell quality was assessed under the microscope and the cells were counted for scRNA-seq library preparation. The listed suspension volumes were used for individual brain regions and were scaled up in case of whole brain or using multiple brain for one sample. For a subset of libraries (see Dataset EV1), a methanol-fixation protocol (Alles et al, 2017) was used prior to library preparation. To fix the cells, they were dissociated, washed and filtered as usual and then resuspended in 200 µl HBSS in a 1.5 ml tube. 800 µl ice-cold 100% methanol was slowly added to the cells while vortexing, after which they were incubated on ice for 30 min and then transferred to −80 °C. The following day the cells were pelleted by centrifugation for 10 min at $1000 \times g$ at 4 °C, washed once with 1 ml PBS + 1U/µl SuperaseIN RNAse Inhibitor (Invitrogen) and resuspended in a final volume of 100 µl HBSS + 1U/µl SuperaseIN. For the papain dissociation, the Papain Dissociation System (Worthington Biochemical Corporation) was used according to the manufacturer's instructions with adjusted volumes for low input material, so that each telencephalon was dissociated in 250 µl papain solution.

## Fluorescence-activated cell sorting

For the samples from Tg[*gfap:GFP*] line, where radial glia carry the transgenic fluorescent label, a sorting step by FACS was performed after dissociation in order to enrich radial glial cells. The cell suspension was resuspended in 1 ml HBSS with 0.01% BSA and labeled with propidium iodide as a dead cell marker (Thermo Fisher, 1:4000 dilution). The GFP(+), PI(−) cells were sorted into a BSA-coated tube and centrifuged for 5 min at $300 \times g$ at 4 °C. The pellet was resuspended in 70 μl HBSS, the cells were counted and quickly loaded on the Chromium Controller.

## Bulk SLAM-seq libraries

For assessment of RNA labeling rates in the adult fish brain, we performed bulk SLAM-seq. A 200 mM 4sU dilution was prepared for injection (170 μl 235.3 mM intermediate 4sU dilution in 10 mM Tris-HCl pH 7.4 + 30 μl Methylene Blue 20 mg/ml) along with a vehicle control (170 μl 10 mM Tris-HCl pH 7.4 + 30 μl Methylene Blue 20 mg/ml). Following a brief anesthesia, the 4sU or vehicle control solution was delivered to the fish either by intraperitoneal (25 μl) or intraventricular injection (3–5 μl). Three biological replicates were performed for each injection. The brains were extracted 6 h after the injection, separated into telencephalon and the rest of the brain, and each part was placed in 500 μl Trizol solution. RNA extraction and library preparation was performed by an adapted CELseq2 protocol as described in Neuschulz et al, (Neuschulz et al, 2023).

## Perturbation experiments and single cell SLAM-seq

To assess the response to a perturbation of the Notch signaling pathway, we treated fish with the γ-secretase inhibitor DAPT (Sigma-Aldrich) as described in Chapouton et al (Chapouton et al, 2010). The fish were incubated in a water bath at a final concentration of 50 μM DAPT (prepared by diluting a 10 mM stock of DAPT in DMSO 1:200 in system water) for 48 h, with a change of water at 24 h. The control fish were incubated in water with 1:200 DMSO dilution.

In order to label nascent transcripts, 3 μl of 200 mM 4sU was delivered to fish brains by intraventricular injection 6 h prior to sample collection. The brains were then collected, and a single-cell suspension was prepared by papain dissociation, with care taken at all steps to reduce light exposure to the sample as much as possible. The final cell suspension was resuspended in 200 μl PBS and filtered through a 40 μm cell strainer into BSA-coated 1.5 ml tubes. The cells were then fixed by slow addition of 800 μl 100% ice-cold methanol while vortexing and left on ice for 45 min. To start the conversion of 4sU, 111 μl of freshly prepared 100 mM iodoacetamide (Sigma-Aldrich) was added to the sample and incubated overnight at room temperature with gentle rotation. The next day the sample was placed on ice for 5 min, after which the cells were centrifuged for 5 min at $1000 \times g$ at 4 °C. The cells were resuspended in 1 ml of quenching buffer (DPBS with 0.1% BSA + 1 U/μl SuperaseIN RNAse inhibitor + 100 mM DTT) and incubated for 5 min at room temperature to block the iodoacetamide. Afterwards the cells were again centrifuged for 5 min at $1000 \times g$ at 4 °C, and the pellet was resuspended in 50 to 100 μl of resuspension buffer (PBS with 0.01% BSA and 0.5 U/μl SuperaseIN + 1 mM DTT). The cells were filtered through a 40 μm cell strainer, checked and counted under the microscope and further processed with 10x Chromium according to the standard scRNA-seq protocol.

## High-throughput lineage tracing based on CRISPR-Cas9

In order to record developmental lineage trees, we further expanded on the CRISPR-Cas9 lineage tracing method as described in Spanjaard et al (Spanjaard et al, 2018). The original method was based on injection of Cas9 and sgRNAs targeting a fluorescent protein transgene, with the repair resulting in small mutations at the Cas9 cut site ('genetic scars') that were used as lineage barcodes. Here, in addition to delivery of Cas9 by injection we used a transgenic line with permanent ubiquitous Cas9 expression (Tg[*ubi:cas9,U6:sgRNA(dTomato),cmlc2:EGFP]*). To establish the line, the dTomato specific target sequence, created by annealing forward (5′- GGACGGCGGCCACT ACCTGGGT-3′) and reverse (5′- CCAGGTAGTGGCCGCCGTC CGA -3′ primers), was cloned into the *pDest(ubi:cas9,U6:sgRNA,cmlc2:EGFP])* plasmid (Ablain et al, 2015) using the BseRI restriction sites. The plasmid was injected into wild-type embryos for Tol2-mediated transgenesis (co-injecting 25 ng/μl Tol2 mRNA with 25 ng/μl plasmid DNA). After screening for *cmlc2*:EGFP expression, the positive F0 founders were raised to adulthood and the offspring with germline integrations were raised as F1 founders. Upon reaching adulthood, the F1 founders were further screened for efficient scar creation in the dTomato sequence, as assessed by a loss of fluorescence in a cross with the *ubi:Zebrabow-M* line which provided the dTomato target gene. Having established a stable line, the lineage tracing experiments were carried out in a cross with the *ubi:Zebrabow-M* line.

In addition to targeting dTomato, we also expanded the number of targets by using the strategy of targeting the 3' untranslated region of highly expressed endogenous genes. This approach leads to improved detection efficiency of lineage barcodes in a flexible way without needing to create new transgenic lines. Two main criteria that were considered for target selection were high expression per cell and uniform expression across different cell types in a previous scRNA-seq zebrafish dataset (Spanjaard et al, 2018). We reasoned that positioning the sgRNAs in untranslated regions of the gene would result in scars that are not harmful to gene function and survival of the fish. This was tested for all target genes before using the sgRNAs in adult fish by injecting the sgRNAs along with Cas9 in embryos and monitoring survival and phenotype until 5 dpf. We did not detect an increase in lethality above baseline levels, and we did not observe any phenotypic changes in response to the injections. The sgRNAs targeting the following genes were selected: *actb1*, *actb2*, *cfl1*, *cirbpb*, *rpl39*, and *ube2e1* (sgRNA sequences for each target are provided in Dataset EV6). For lineage tracing experiments, an equimolar mixture of the sgRNAs at the final concentration of 250 ng/μl was co-injected along with Cas9 protein (NEB, final concentration 350 ng/μl) into 1-cell stage embryos in 2 nl droplets. In order to screen for embryos with successful gene editing events, a sgRNA targeting the main ORF of *tyrosinase* gene (Yin et al, 2015) was added to the mix, and the partial loss of pigmentation resulting from successful editing of tyrosinase was used as a selection criterion for raising embryos to adulthood.

## Single-cell library preparation and sequencing

The single cell suspensions were loaded on the Chromium Controller (10X Genomics) aiming for the highest possible number of captured cells. The libraries were prepared with the Single Cell 3' and 5' kits (kit version used to prepare each library can be seen in Dataset EV1).

The standard transcriptome libraries were prepared according to the manufacturer's instructions. For targeted libraries of scars used for lineage tracing, the remaining cDNA was used to set up individual PCR reactions for each target. In an optional nested PCR used for some targets, the target gene of interest was first enriched. In the next round of PCR a primer with a partial Illumina adapter sequence was used along with the 10× primer to specifically amplify the sgRNA target site. The list of target-specific primers can be seen in Dataset EV6. In the final library PCR, the amplicon was indexed with the standard library primers from the 10x Genomics kit, which enabled the targeted libraries to be sequenced with and demultiplexed along with the transcriptome. After quality control of libraries on Bionalyzer on Tapestation (Agilent), the libraries were sequenced on one of the Illumina sequencing platforms.

## Pre-processing and clustering of scRNA-seq data

The *fastq* files of the standard transcriptome libraries were mapped with CellRanger 6.1.1. to a zebrafish genome prepared from GRCz11, release 92. We detected a variable amount of mitochondrial reads between samples (see Dataset EV1), so in order to reduce batch effects, the gene expression matrices were first processed with the SoupX tool for removing ambient RNA contamination (Young and Behjati, 2020). In addition, this allowed us to account for potentially different levels and compositions of ambient background RNA in samples treated with FACS vs. non-FACS-treated samples.

The corrected matrices were then processed with the doublet-prediction tool scDblFinder (Germain et al, 2022) in order to identify putative doublet cells. The flagged doublets were removed and downstream analysis was done in Seurat v4.2.0. Cells were filtered for quality based on gene and transcript counts as well as percent of mitochondrial genes (200–2500 genes, up to 10,000 UMIs and up to 25% mitochondrial genes per cell barcode). The clustering was done in two steps. Initially, the full datasets were clustered together to identify major cell types based on the top 3000 most variable genes in the dataset. Number of genes, UMIs and mitochondrial content were regressed out during data scaling. The first 28 principal components were included for UMAP clustering. Main cell types were identified based on known markers found within the calculated marker genes for each cluster (see Dataset EV2).

Subsequently, the four largest cell type groups (neurons, radial glia, oligodendroglia and immune cells) were separately analyzed based on the top 1000 most variable genes in each dataset, with regression of confounding variables (gene and transcripts counts, percentage of mitochondrial reads, library of origin) during data scaling. The resulting clusters were individually inspected in two steps before assigning subtypes. First, the sample composition of each cluster was calculated and the clusters in which more than 90% of the cells of the clusters originated from a single library were excluded as likely influenced by batch effect. Second, following the identification of differentially expressed genes for each cluster, they were compared across clusters and the clusters that could not be individually defined by a unique combination of 1–3 specific markers were merged with the most similar larger cluster. This resulted in lists of cell subtypes that were characterized by a considerable number of differentially expressed marker genes (Dataset EV3). The final clusters were annotated based on a biologically known function, if possible, or a characteristic marker gene. The full list of identified cell types and selected marker genes can be found in the Dataset EV2, and a full list of marker genes for all radial glia subtypes is provided in Dataset EV3.

## Integration with larval datasets

In order to examine the developmental time point at which the detected radial glia subtypes may emerge, we compared our dataset of adult brain cells to a published larval dataset of the zebrafish brain (Raj et al, 2020). The dataset was downloaded from the provided GEO accession number as a clustered Seurat object. For the analysis we considered only time points between 20 hpf and the last timepoint of 15 dpf, since at very early stages preceding 20 hpf the cell types identified in the publication do not share enough marker similarity with our dataset. We selected all cells classified as radial glia, glial progenitors, neural progenitors, proliferating progenitors and differentiating cells. The cells were integrated with the radial glia of the adult brain using Seurat CCA function with 3000 variable genes and Louvain clusters were identified in the shared embedding. To quantify the similarity of adult cell types with the developmental cell types, for each of the adult cell types we took into account all the new clusters in which a specific adult cell type was found. The cells from all the clusters containing that adult cell type were combined, weighted by adult contribution (e.g. with a factor of 0.25 if 25% of the adult cells were found in that cluster) and a normalized percentage of the developmental stage of origin of these cells was calculated (Fig. 3A).

## GO term, KEGG pathway and Reactome pathway analysis

Pathway analysis was conducted for all marker genes with an adjusted *p*-value of <= 0.01 for each radial glia subcluster. To find significantly enriched terms in all three databases, we used the gost function from the gprofiler2 R-package (v. 0.2.2) (Raudvere et al, 2019; Kolberg et al, 2020) with the organism set to "drerio" and correction method set to "fdr". Only terms with a *p*-value of 0.01 or lower were included as significant. In addition, we required relevant terms to be present in no more than 2 subclusters, to filter out general radial glia related terms.

## Disease model analysis

Genes that were implicated in specific diseases or abnormalities of the nervous system were obtained from the Mouse Genome Informatics database (https://www.informatics.jax.org/) of the Jackson Laboratory. We selected genes used in mouse models for the following 15 mammalian phenotype ontology annotations related to astrocytes (mammalian homologues of adult zebrafish radial glia), neural stem cells and radial glia: decreased astrocyte number, astrocytosis, increased astrocyte size, radial glia endfoot detachment, decreased radial glia cell number, increased radial glial cell number, ectopic Bergmann glia cells, abnormal neuronal stem cell morphology, impaired neuron differentiation, premature neuronal precursor differentiation, abnormal embryonic neuroepithelial cell proliferation, abnormal astrocyte physiology, abnormal neuronal stem cell physiology, enhanced central nervous system regeneration and impaired central nervous system regeneration. This resulted in a list of 173 unique mouse genes, which were translated to zebrafish orthologues with the orthology search function of g:Profiler (https://biit.cs.ut.ee/gprofiler/orth)

(Dataset EV5). Since not all the obtained genes were expressed in the dataset highly enough to notice meaningful patterns, they were filtered based on average expression in radial glia cells for genes with average expression of 1 in at least one radial glia subtype, resulting in a final list of 54 genes (Appendix Fig. S2).

## RNAscope

Brains for RNAscope were isolated from the 18 months old transgenic gfap:GFP line and fixed for 3 h at 4 °C in 4% paraformaldehyde (PFA) in phosphate-buffered saline (PBS). Fixed brains were cryoprotected in increasing sucrose solutions of 10, 20, and 30% in PBS for 24 h at 4 °C each, embedded in NEG-50 Frozen Section Medium (Epredia), cryosectioned in 20 μm sections that were mounted on SuperFrost-Plus slides (Epredia). In situ hybridization was performed according to ACD RNAscope™ Multiplex Fluorescent Reagent Kit v2 for fixed-frozen tissue (UM 323100) manufacturer protocol using probes designed by ACD. Fluorophores were diluted 1:1500. For cell immunolabeling of radial glial cells, anti-GFP (1:750, Abcam) was used. The primary antibody was detected by Alexa Fluor 488 (1:1000, Invitrogen). All sections were embedded in Aqua Polymount (Polyscience).

The RNAscope signal for selected genes was quantified using Fiji software (version 2.9.0/1.53t). The RNAscope signal was measured in the section area containing radial glia cells. The gfap:GFP signal was used to create the area for quantification. After thresholding the GFP signal, the homogeneous area was generated using Fiji "dilatation" and "fill hole" functions. After extracting the region of interest (ROI), this ROI was applied to the thresholded RNAscope signal and the fraction of the ROI covered by the RNAscope signal (Fraction of Area) was measured.

## Mapping and filtering of endogenous scar libraries

The targeted lineage tracing libraries were aligned using bwa mem3 (v.0.7.12) to individual references of endogenous genes used in the lineage tracing experiment (*actb1*, *actb2*, *cfl1*, *cirbpb*, *rpl39,* and *ube2e1*). The reads that were associated with valid barcodes (defined as barcodes passing quality filtering in the transcriptome analysis) and started with the gene-specific primer sequence were retained and truncated to 75 nucleotides. Subsequently, the scars were filtered with a custom pipeline to eliminate sequences that originate from PCR or sequencing errors. The underlying assumption for the filtering was that for each endogenous gene, given that it can be edited in the two alleles, up to two valid sequences (wildtype or scar) can be present in each cell. For each target gene, the number of sequences that contributed to the top 80% of transcripts associated with each barcode was counted, and in cases where more than two sequences were present the specific target was flagged for removal. In cases where more than one target was flagged, the cell itself was removed from the dataset as a putative doublet. The cells and scars that passed this filtering step were used as input for the clone identification pipeline.

## Method for clone identification from scars in endogenous genes

Clone determination starts out with a separate analysis of each target site. A sequence-ID of 40 bases around the target site is used

as the scar-ID. This has the advantage of distinguishing distinct scars that have the same structure and hence the same CIGAR, while excluding the distinction of scars purely based on sequencing errors that are far away from the target site.

We now remove sequence-IDs that are only detected in one cell and select all cells that have two distinct sequence-IDs (having one wildtype-ID and one scar-ID is allowed). Using this set of cells, we determine all sequence-ID pairs and remove combinations that occur in three or fewer cells. Then, we go through each observed sequence-ID pair. For each sequence-ID of the pair, we calculate which fraction of the total observations of that ID occur in this specific sequence-ID combination. If at least 80% of observations of one or both sequence-IDs are found in this specific combination, we call this a true pair and keep it for further analysis. The reasoning here is that the two sequence-IDs in a pair have either been created at a similar timepoint, meaning that they should always occur together, or that one scar has been created way earlier than the other, meaning that the first sequence-ID ("parent scar") potentially appears in many combinations, while the second sequence-ID ("child scar") appears exclusively in this combination. If both sequence-IDs occur in more than one combination at substantial levels, this means that at least one of the scars has likely been created multiple times and can't be used for sequence-ID-pair identification. The cut-off is set at 80% to account for occasional mis-assignment of sequence-IDs because of, e.g., chimeric reads.

The "child scar", or in some cases both scars, can be used as the clone-defining sequence, which allows us to now place cells, in which only one sequence-ID could be detected, into existing clones. Each cell, in which one single clone-defining sequence-ID was detected, will be placed into the appropriate clone. The wildtype-sequence as well as "parent scars", which are detected in many combinations, are excluded from serving as the clone-defining sequence-ID.

Once we have defined clones on a per-target basis, we merge the information on multiple targets. This is done in a pairwise manner for all possible combinations of two targets and follows the same steps as the per-target clone identification. Briefly, we remove combinations of clone-IDs that only occur in one cell and only keep cells that have been assigned a clone-ID for both targets of interest. Only clone-ID-combinations that appear in three or more cells are kept. For each clone-ID-pair, the fraction of the total observations of each clone-ID that can be attributed to this specific combination is calculated. We only keep combinations, which account for at least 80% of the total observations of one of the scars. Again, the clone-ID that occurs more consistently in a specific pair, will be set as the clone-defining ID. If both clone-IDs occur almost exclusively in that pair, they are both considered to be clone-defining IDs. Now, cells that have only been assigned to a clone for one of the targets, more specifically a clone-ID that has been determined to be clone-defining, can be placed into double-target clones. We now gather the information of all cells that have been placed into a clone defined by any two targets. Duplicate information for cells that have been assigned to a clone for more than one pair of targets is removed. This set of cells can now be used for visual assessment of, e.g., cell type-distribution across clones by hierarchical clustering visualized in a heatmap. In total, the pipeline achieves the identification of clone-defining combinations of sequences and the placement of cells into the clones. Even cells, in which only one scar on each of two targets could be detected, can be placed into a

clone. The pipeline ignores the possibility that both alleles of a target carry the same sequence ID. Cells, in which the same scar was created twice, as well as cells that have two wildtype-alleles are being excluded from the analysis.

## Quantification of bulk metabolic labeling rate

The bulk SLAM-seq libraries were analyzed as described in Neuschulz et al (Neuschulz et al, 2023). Briefly, the fastq files were mapped with STAR v2.7.3a to a zebrafish genome reference GRCz11, release 95, with reference information on substitutions (MD tag) included. The information on substitution type, location, and sequencing quality was aggregated in an additional tag (MT). After eliminating substitutions with low sequencing quality scores (<Q20), the frequency of substitutions for each sample was plotted in order to to compare the rate of metabolic labeling (T to C substitution) with the background substitution rate (Fig. 5B).

## Single-cell RNA velocity determination

We used two different quantities to calculate single-cell RNA velocity: transcripts that (partially) align to intronic regions of genes, and T-to-C mutations in transcripts induced through metabolic labeling. All datasets contain intronic reads but only scSLAM-seq datasets (Notch inhibition and control datasets, Fig. 5C) include metabolic labeling. Datasets with intronic reads only were aligned to the zebrafish reference using Cellranger 6.1.1. The output was fed into the Velocyto (v0.17.17) (La Manno et al, 2018) wrapper run10x for 10x Chromium samples which creates count matrices for unspliced and spliced molecules. Consequently, the samples were preprocessed and clustered with scVelo's (v0.2.3) (Bergen et al, 2020) implementation of the Leiden clustering. Furthermore, the relative positioning of cells was explored with scanpy's (v1.7.1) (Wolf et al, 2018) umap implementation. We obtained the RNA velocity for each cell by analyzing gene expression changes with scVelo. This step resulted in velocity matrices, that describe cell-to-cell transitions, which were further investigated by examining the proportions of transition probabilities between and within clusters of each cell. For this step, the maximal probability for each cell was extracted and transitions normalized per cluster.

After this, we analyzed separate datasets using CellRank (Lange et al, 2022) (v1.5.2): we used Generalized Perron Cluster Cluster Analysis (GPCCA) on a connectivity kernel to compute the 17 highest-eigenvalue macrostates. We used these macrostates as terminal states in a GPCCA estimator on a mixed velocity and connectivity kernel (0.8 velocity and 0.2 connectivity) and computed aggregate absorption probabilities from the predefined cell types to the calculated terminal states.

Datasets that combine intronic reads and metabolic labeling were preprocessed using dynast (v1.0.1) (https://github.com/aristoteleo/dynast-release). Sequencing data was aligned to a zebrafish transcriptome created with dynast from GRCz11, release 92. We then consecutively used the align, count and estimate commands, providing a file of whitelisted barcodes we had previously determined (barcodes that pass quality control filters as part of the Seurat analysis). Next, we analyzed the data using a combination of Dynamo (Qiu et al, 2022) (v1.1.0), scanpy (Wolf et al, 2018) (v1.9.1) and scVelo (Bergen et al, 2020) (v0.2.4). For the

analysis of the single datasets and the analysis of all datasets merged together, we used the same procedure. We focused on the neuronal and radial glia cells and preprocessed using Dynamo's recipe-monocle function. We then determined the dynamics using the sci-fate method and a stochastic model. After using scanpy's neighbor-determination and umap algorithm with default parameters, we plotted the data using scVelo's embedding_stream functionality.

## Data availability

The datasets and computer code produced in this study are available in the following databases: single-cell RNA-seq: Gene Expression Omnibus GSE246714, reviewer access token avabaiksv-jiftud. Custom code: Github (https://github.com/nimitic/radial-glia).

## Peer review information

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

## Acknowledgements

We acknowledge support by MDC/BIMSB core facilities (zebrafish, genomics, bioinformatics), and we thank J. Richter for help with zebrafish experiments. Work in JPJ's laboratory was funded by a European Research Council Starting Grant (ERC-StG 715361), a Consolidator Grant (ERC-CoG 101043364), and a Helmholtz AI grant (ZT-I-PF-5-54). AN was supported by a PhD fellowship from *Studienstiftung des deutschen Volkes*.

## Author contributions

**Nina Mitic**: Conceptualization; Data curation; Formal analysis; Investigation; Visualization; Methodology; Writing—original draft. **Anika Neuschulz**: Software; Formal analysis; Investigation; Visualization; Methodology; Writing—review and editing. **Bastiaan Spanjaard**: Software; Formal analysis; Visualization; Methodology. **Julia Schneider**: Validation; Investigation. **Nora Fresmann**: Formal analysis; Investigation; Visualization; Methodology. **Klara Tereza Novoselc**: Investigation; Methodology. **Taraneh Strunk**: Software; Formal analysis. **Lisa Münster**: Methodology. **Pedro Olivares**-Chauvet: Software; Methodology. **Jovica Ninkovic**: Supervision; Funding acquisition; Writing—original draft; Writing—review and editing. **Jan Philipp Junker**: Conceptualization; Supervision; Funding acquisition; Writing—original draft; Project administration; Writing—review and editing.

## Funding

## Disclosure and competing interests statement

The authors declare no competing interests.

