## [Peer Review File · Molecular Systems Biology]

Dissecting the spatiotemporal diversity of adult neural stem cells

Jan Philipp Junker, Nina Mitic, Anika Neuschulz, Bastiaan Spanjaard, Julia Schneider, Nora Fresmann, Klara Novoselc, Taraneh Strunk, Lisa Münster, Pedro Olivares, and Jovica Ninkovic

Corresponding author(s): Jan Philipp Junker (janphilipp.junker@mdc-berlin.de)

Review Timeline:

Submission Date:	4th May 23
Editorial Decision:	30th May 23
Revision Received:	1st Nov 23
Editorial Decision:	19th Dec 23
Revision Received:	31st Jan 24
Accepted:	1st Feb 24

Editor: Maria Polychronidou

Transaction Report:

30th May 2023

Manuscript Number: MSB-2023-11757

Title: Dissecting the spatiotemporal diversity of adult neural stem cells

Dear Philipp,

Thank you again for submitting your work to Molecular Systems Biology. We have now heard back from the three reviewers who agreed to evaluate your study. As you will see below, the reviewers raise substantial concerns, which preclude the publication of the study in its current form.

While reviewer #1 is more supportive, reviewers #2 and #3 point out that the study remains preliminary. They mention that several conclusions need to be better supported by additional analyses and raise significant concerns regarding the data analysis and presentation. Nevertheless, the reviewers acknowledge that the presented data and findings are a potentially relevant contribution to the field and they provide constructive suggestions on how the study could be revised and extended in order to address their concerns. Taken together and considering that the reviewers acknowledge the overall relevance, we would like to offer you a chance to address the issues raised in a major revision. Without repeating all the points listed below, some of the more fundamental issues are the following:

- Follow up analyses are required to better support the main conclusions and to enhance the level of biological insight provided by the study. The reviewers provide suggestions in this regard, and this is an important point to address during revision.
- Further details regarding the methodology, the datasets and the performed analyses need to be provided. Moreover, the data documentation and presentation need to be improved.

All issues raised by the reviewers need to be satisfactorily addressed. As you may already know, our editorial policy allows in principle a single round of major revision, so it is essential to provide responses to the reviewers' comments that are as complete as possible. I understand that the required revisions are substantive. Please feel free to contact me in case you would like to discuss in further detail any of the issues raised or if you would like to share your revision plan with me. I would be happy to schedule a call.

On a more editorial level, we would ask you to address the following points:

- Please provide a .doc version of the manuscript text (including legends for the main figures) and individual production quality figure files for the main Figures (one file per figure).
- Please include 5 keywords.
- We have replaced Supplementary Information by the Expanded View (EV format). In this case, all additional figures can be included in a PDF called Appendix. Appendix figures should be labeled and called out as: "Appendix Figure S1, Appendix Figure S2... Appendix Table S1..." etc. Each legend should be below the corresponding Figure/Table in the Appendix. Please include a Table of Contents in the beginning of the Appendix. For detailed instructions regarding expanded view please refer to our Author Guidelines: .
- Supplementary Tables 1-3 should be provided as Datasets EV1-EV3. Please provide one file per dataset and include a description of the dataset in the .xls file as a separate tab.
- Please provide a "standfirst text" summarizing the study in one or two sentences (approximately 250 characters), three to four "bullet points" highlighting the main findings and a "synopsis image" (exactly 550px width and max 400px height, jpeg format) to highlight the paper on our homepage.
- All Materials and Methods need to be described in the main text. We would encourage you to use 'Structured Methods', our new Materials and Methods format. According to this format, the Material and Methods section should include a Reagents and Tools Table (listing key reagents, experimental models, software and relevant equipment and including their sources and relevant identifiers) followed by a Methods and Protocols section in which we encourage the authors to describe their methods using a step-by-step protocol format with bullet points, to facilitate the adoption of the methodologies across labs. More information on how to adhere to this format as well as downloadable templates (.doc or .xls) for the Reagents and Tools Table can be found in our author guidelines: . An example of a Method paper with Structured Methods can be found here:
- Please include a "Disclosure and Competing Interests Statement " in the main text.

- Please include a Data availability section describing how the data, code etc. have been made available. This section needs to be formatted according to the example below:

The datasets and computer code produced in this study are available in the following databases:

- Chip-Seq data: Gene Expression Omnibus GSE46748 (<https://www.ncbi.nlm.nih.gov/geo/query/acc.cgi?acc=GSE46748>)

- Modeling computer scripts: GitHub (<https://github.com/SysBioChalmers/GECKO/releases/tag/v1.0>)

- [data type]: [full name of the resource] [accession number/identifier] ([doi or URL or identifiers.org/DATABASE:ACCESSION])

- The References should be formatted according to the Molecular Systems Biology reference style (i.e., ordered alphabetically and listing the first 10 authors followed by et al.)

- For data quantification: please specify the name of the statistical test used to generate error bars and P values, the number (n) of independent experiments (specify technical or biological replicates) underlying each data point and the test used to calculate p-values in each figure legend. The figure legends should contain a basic description of n, P and the test applied. Graphs must include a description of the bars and the error bars (s.d., s.e.m.).

- When you resubmit your manuscript, please download our CHECKLIST (<https://bit.ly/EMBOPressAuthorChecklist>) and include the completed form in your submission.

Please note that the Author Checklist will be published alongside the paper as part of the transparent process (<https://www.embopress.org/page/journal/17444292/authorguide#transparentprocess>).

If you feel you can satisfactorily deal with these points and those listed by the referees, you may wish to submit a revised version of your manuscript. Please attach a covering letter giving details of the way in which you have handled each of the points raised by the referees. A revised manuscript will be once again subject to review and you probably understand that we can give you no guarantee at this stage that the eventual outcome will be favorable.

Kind regards,

Maria

Maria Polychronidou, PhD
Senior Editor
Molecular Systems Biology

We realize that it is difficult to revise to a specific deadline. In the interest of protecting the conceptual advance provided by the work, we recommend a revision within 3 months (28th Aug 2023). Please discuss the revision progress ahead of this time with the editor if you require more time to complete the revisions. Use the link below to submit your revision:

IMPORTANT: When you send your revision, we will require the following items:

1. the manuscript text in LaTeX, RTF or MS Word format
 2. a letter with a detailed description of the changes made in response to the referees. Please specify clearly the exact places in the text (pages and paragraphs) where each change has been made in response to each specific comment given
 3. three to four 'bullet points' highlighting the main findings of your study
 4. a short 'blurb' text summarizing in two sentences the study (max. 250 characters)
 5. a 'thumbnail image' (550px width and max 400px height, Illustrator, PowerPoint or jpeg format), which can be used as 'visual title' for the synopsis section of your paper.
 6. Please include an author contributions statement after the Acknowledgements section (see <https://www.embopress.org/page/journal/17444292/authorguide>)
 7. Please complete the CHECKLIST available at (<https://bit.ly/EMBOPressAuthorChecklist>).
- Please note that the Author Checklist will be published alongside the paper as part of the transparent process (<https://www.embopress.org/page/journal/17444292/authorguide#transparentprocess>).
8. When assembling figures, please refer to our figure preparation guideline in order to ensure proper formatting and readability in print as well as on screen:

See also figure legend guidelines: <https://www.embopress.org/page/journal/17444292/authorguide#figureformat>

9. Please note that corresponding authors are required to supply an ORCID ID for their name upon submission of a revised manuscript (EMBO Press signed a joint statement to encourage ORCID adoption).

(<https://www.embopress.org/page/journal/17444292/authorguide#editorialprocess>)

Currently, our records indicate that the ORCID for your account is 0000-0002-2826-8290.

Link Not Available

The system will prompt you to fill in your funding and payment information. This will allow Wiley to send you a quote for the article processing charge (APC) in case of acceptance. This quote takes into account any reduction or fee waivers that you may be eligible for. Authors do not need to pay any fees before their manuscript is accepted and transferred to the publisher.

EMBO Press participates in many Publish and Read agreements that allow authors to publish Open Access with reduced/no publication charges. Check your eligibility: <https://authorservices.wiley.com/author-resources/Journal-Authors/open-access/affiliation-policies-payments/index.html>

*** PLEASE NOTE *** As part of the EMBO Press transparent editorial process initiative (see our Editorial at <https://dx.doi.org/10.1038/msb.2010.72>), Molecular Systems Biology publishes online a Review Process File with each accepted manuscripts. This file will be published in conjunction with your paper and will include the anonymous referee reports, your point-by-point response and all pertinent correspondence relating to the manuscript. If you do NOT want this File to be published, please inform the editorial office at msb@embo.org within 14 days upon receipt of the present letter.

Reviewer #1:

In this manuscript Mitic et al presents powered single cell transcriptomics and lineage tracing modalities in adult zebrafish brain, a prime regeneration model in the nervous system of vertebrates. Authors sequenced around 100,000 cells throughout the zebrafish brain and identified various cell types with their canonical markers. Some of the cell types were global whereas some were regionally specified, signifying the spatial organisation of the stem cell niches and neurogenic ability. Integrating their dataset to developmental datasets of the zebrafish brain, authors find that radial glia diversity might arise in the embryonic development, which was supported by other findings in the literature. To test the regional origins of the radial glia, authors used parallel gene editing with single-guide RNAs. This approach is a clever way to generate "genetic scars" that can be later identified as lineage barcodes. This strategy suggested that glia might be regionalised within their own niche due to extrinsic factors of that particular brain region. This hypothesis is also supported by velocity studies where authors propose that global glial states could be linked to a neurogenic ability while regional states could be defining structural/other physiological functions. An interesting finding is that snap25+ glia could transdifferentiate into neurons without proliferation while her4+ glia is a proliferative stem cell indicator. To determine the directionality of the cellular physiologies, authors present for the first time scSLAM technique implemented to a live vertebrate brain. Authors state the limitations of the study as dissection of the brain to cause a large spatial region that reduces the cellular resolution mapping to brain microregions, and lineage analyses being limited to the early development.

The study is an important contribution to our understanding of the diversity and organization of neural stem cells in the adult zebrafish brain, which has a high regenerative capacity. The single-cell transcriptomics, lineage tracing, and RNA metabolic labeling techniques to analyze the regulation of neural stem cells in different brain regions over time are valuable technical demonstrations for the field. The documentation of a wide range of neural stem cell states, some of which were specific to certain brain regions while others were found throughout the brain, is important for regenerative medicine. The proposals that the global stem cell states were associated with neurogenic differentiation, with different states being involved in either proliferative or non-proliferative differentiation can be important for clinical ramifications for developmental disorders. The findings have the potential to shed light on the principles of adult stem cell organization and to provide insights into manipulating specific subtypes of neural stem cells for functional purposes.

The writing language in the manuscript appears to be clear and concise. The authors use scientific terminology and technical language appropriate for a research paper. The text is generally well-structured, making it easy to follow the flow of ideas and understand the research methods and results.

In overall, the manuscript is suitable for publication pending on addressing the major comments below.

There are a few inconsistencies or unclear statements in the text that could be improved:

In the Abstract, it states, "we combined single-cell transcriptomics of dissected brain regions with massively parallel lineage tracing and in vivo RNA metabolic labeling to analyze regulation of neural stem cells in space and time." However, in the Introduction, it mentions the use of single-cell RNA-seq and lineage tracing, but does not specifically mention in vivo RNA

metabolic labeling. The specific details of the methods used may not be fully consistent between the Abstract and Introduction. In the Results section, it mentions the identification of 23 neuronal subtypes and 18 radial glia subtypes. However, in the Introduction, it states that there were 23 identified neuronal subtypes and 18 subtypes of immune cells, but does not explicitly mention the number of radial glia subtypes. This discrepancy could be clarified.

The term "global radial glia" is used to describe certain radial glia states found across all four brain regions. However, in the context of lineage tracing and developmental origin, it is mentioned that there is no early developmental lineage split between global and regional radial glia. This seems contradictory, as the term "global" suggests a distinction in lineage or origin. The concept of "global" radial glia states could be better defined or explained to avoid confusion.

Overall, while these inconsistencies do not significantly affect the main findings of the study, clarifying and providing more specific information in these areas would enhance the coherence of the text.

The authors may try to link the identified cellular markers to more referencing and literature as there are many publications that document developmental, homeostatic and pathology-induced radial glial gene expression profiles. Could it be possible to overlap the lineage transitions or trajectory node markers to external references (e.g. gene X in transition from RG subtype A to B in comparison to a disease model or a specific perturbation)? The nature of the advance in this manuscript as is seems to be technical and sample power-related rather than conceptual. Additionally, the advance compared to previous knowledge in the zebrafish field can be better delineated with more rigorous referencing.

Can authors include GO term and KEGG pathways analyses to the manuscript to better define the subtypes for their molecular programs? As is, the manuscript is depicting only gene expressions that may not be fully explaining the physiological relevance of these cell subtypes. Additionally, rigorous discussion on the identified GO terms would enhance the quality of the paper.

There are several interesting hypothesis in this paper but they should be supported by secondary validations (pathway analyses etc). Authors claim the presence of regional specificities of the RGs, but they do not provide any immunostainings or other type of validation data to show that regional cells with specific markers do exist in those regions.

The authors use gfap:GFP transgenic animal that marks the astroglia-like cells in the zebrafish brain. This line does not mark all types of radial glia. Can a quality control step be implemented showing the colocalization of gfap:GFP cells with other glial markers to have a quality measure?

The power of the studies are fantastic. However, the methodological details can be explained more and documentation of the sample characteristics can be significantly improved. The corresponding author is a leading expert on the topic, but the reviewer GEO link was not provided. The reviewers cannot comment on the data content and quality.

Reviewer #2:

Comments on Paper Mitic et al., Dissecting the spatiotemporal diversity of adult neural stem cells.

General comments

The authors based all of their conclusions on sequencing data to which they:

- a. do not show any plots to indicate the resolution and the quality of sequencing (example of such plots would be ones showing mitochondrial genes distribution).
- b. do not indicate how many replicates are used, does each biological replicate consist of only one zebrafish brain or are they pooled (and how many are pooled)
- c. show no validation at all of any sequencing data on the clustering of different regions. Such validations are essential and could have been easily performed with immunostainings or RNAscope for example.

Specific Figure comments

Fig.1

It is not clear to the reviewer how the clustering was performed. A more detailed description on how the clustering was done is needed. Just having 1-3 genes is not strong enough to claim that these are distinct types (eg. in several GABAergic clusters, the difference is based on only one marker expression, which could be stochastic expression.)

There is inconsistency in switching between the terms cell types, cell clusters and cell state throughout the text, which makes it difficult to understand and follow.

Fig.2

It is not clear if all sorted cells were integrated or only radial glial cells.

The authors mention using a threshold of 70% without any explanation on how this threshold was determined.

Fig.3

The reviewer found this figure very hard to follow and understand

It is not clear how one clone can be considered regional and global at the same time!

There is no sequencing data of the clones recovered shown. How do the authors control for the mutations they get with the scarring method? Relying only on the lethality of embryos does not make sense at all, as some baseline lethality is expected anyway when injecting anything into zebrafish embryos at 1-cell stage.

Fig.4

The reviewer found it hard to 'clearly distinguish' the trajectories in Fig 4A as the authors describe in the text.

Error bars are not clear at all and seem to be shifted/moved

The authors indicate in the text that intron/exon based RNA velocity has its caveats and they even cite several papers discussing these caveats at different steps while performing RNA velocity. To remedy this, the authors employ metabolic labelling of nascent RNAs. However the papers cited by the authors point to much deeper problems in RNA velocity pipelines than just the use of introns and exons. For example the way that data is processed by RNA velocity methods is a large problem. As a consequence, Gorin et al. advise against normalisation, imputation and most things that make up RNA velocity pipelines. As a consequence, I think that it is unlikely that using SLAMseq will remedy the problems in how RNA velocity data is analysed. Moreover, how the authors perform RNA velocity analysis is not indicated at all.

Fig.5

Plots in this figure are not explained in the text

Some validation that the notch inhibition treatment is actually working is lacking.

Reviewer #3:

The manuscript entitled "Dissecting the spatiotemporal diversity of adult neural stem cells" by Mitic et al. describes the diversity of radial glia cell states in the adult zebrafish brain. This work is mainly accomplished by use of single-cell RNA sequencing where the authors profiled cells from microdissections of four major brain regions (telencephalon, diencephalon, mesencephalon, and rhombencephalon). The authors present a substantial dataset of 100,000+ cells from zebrafish brains from which they describe several radial glia (RGs) sub-clusters. They classify RGs based on their regional distribution and gene expression profiles. These analyses revealed two spatial distributions of RGs, those that are mostly confined to one region ("regional") and those that are found in multiple regions ("global"). Additionally, the authors reveal several sub-clusters of RGs based on gene expression, for example proliferating RGs, snap25 expressing RGs, and her4 expressing RGs. Using an enhanced version of their previously published method for lineage tracing in zebrafish, the authors show that regional and global RGs are not from different lineages and instead clonal families contain both global and regional RGs. Additionally, the authors use SLAM, scSLAMseq, and Notch inhibition experiments to find that different gene expression based clusters of RGs have different transition probabilities to other cell types/states.

While I believe the data presented and analysis thereof will be beneficial to the community I do have major concerns. One general concern is that the data analysis and data presentation appear preliminary and partially confusing. More work needs to go into extracting information from the data and presenting it in a more clear and concise way. A further major concern is the lack of validation (spatial analysis, staining) of cell populations identified using single-cell transcriptomics. Below I am listing my questions and concerns in more detail:

The authors present an impressive amount of single-cell data across the zebrafish brain; however, it would be beneficial for the authors to include some spatial information on how different clusters across the different brain regions are arranged within the tissue. Spatial information would be particularly of interest for the sub-clusters of radial glia cells, including the small group of neuroepithelial cells or the enkur+ cluster which is speculated to be lining the ventricle (do you find this cluster in the ventricles of other regions outside the telencephalon?). How do subclusters of radial glia relate in space to each other even within a given region?

I could not find the numbers of cells belonging to each major cell type (neurons, radial glia, immune cells, etc.) or to the subtypes of each cell type. These numbers should be stated either in the text or in the figures (n=xx). Additionally, it would be good to know how many cells are contributed from which experiments (unbiased cell profiling vs FACS enriched experiments, etc.). Further, please add information regarding how many replicates are used for these experiments.

Figure 2 - please add an indicator of which clusters are global or regional. This would make it easier for the reader to understand the different sub-clusters that are highlighted.

I understand the authors use SoupX to remove "contaminating" mRNAs. Do the authors identify why this is necessary or which genes they find problematic? Moreover, the authors identify one sub-cluster of radial glia with snap25+ expression, could this be from high ambient RNA from neurons, RG-neuron doublets or is this true signal from radial glia cells? This cluster is of particular interest in the subsequent analysis using scSLAMseq thus the importance of showcasing the "true" signal (for example with

spatial methods) of snap25 would be critical.

Could the authors explain why the diencephalon was the least accurate of the dissections and possibly add this rationale to the sentence they already present in their text or to the methods?

Figure 2A is hard to follow, for example the authors highlight Bergmann glia (cluster 6) and while it is described in the text as Bergmann glia the figure still calls it RG prss35+. The authors should make these annotations more consistent and the figure easier for the reader to navigate.

Figure 3A - authors should avoid pie charts; it is hard to understand the radial glia sub-cluster or the integrated data cluster composition across the different timepoints which makes it hard to interpret the authors findings.

The authors state that radial glia diversity is already established at embryonic and early larval stages, but the UMAP in Figure 3A appears to have many cells that appear primarily from the adult. This part of the analysis could use some more exploration. For instance, an additional supplemental figure that shows why earlier larval timepoints were excluded or if there are radial glia cell clusters from the integrated data that are primarily from adult. Are there any notable differentially expressed genes between adult and larval stages in clusters that are more equal in their cluster contribution (example: RG foxn4+ or the NE cluster)

Figure 3C - this is hard to interpret; labeling the clones (1, 2, 3, ...) could add the reader. Regarding the clone bar annotation, what does it mean when cells are not in a clone bar? Are these single cells with no family? If this is true could this figure be in the supplement and instead highlight the clonal families in the main figure (highlight the sizes of clonal families and the number of cells belonging to different annotation groups like cell type and region)? Can the color choices for Figure 3 and Figure S3C be adjusted to aid the reader with the color bars (particularly for the cell type annotations).

Figure 4B- The pie charts are difficult to understand, the percentages for the dominant cell type that a cell transitions to could be labeled as well as a legend for the cell type annotations.

Figure 4C - RG her4++ appears to transition to the proliferating cells but is also a high amount of cells to neurons_5. This probability actually appears higher than the snap25+ RGs that transition to neurons (which have probabilities to other snap25+ and id2b+ RG clusters). Generally, the plots in 4B and 4C are difficult to understand the author's analysis and conclusions.

Figure 5 figure legend - The reference Fig S5C, however this figure does not exist.

Figure 5D - this panel is hard to interpret. The variation in the fraction of snap25+ RGs involved in in direction differentiation is not clear from the two UMAPs of the treatment and control.

When inhibiting Notch and thus increasing the amount of direct differentiation, is there an observable difference in the neuronal output sub-clusters, are particular neuronal populations associated with direct neurogenesis?

Reviewer #1:

In this manuscript Mitic et al presents powered single cell transcriptomics and lineage tracing modalities in adult zebrafish brain, a prime regeneration model in the nervous system of vertebrates. Authors sequenced around 100,000 cells throughout the zebrafish brain and identified various cell types with their canonical markers. Some of the cell types were global whereas some were regionally specified, signifying the spatial organisation of the stem cell niches and neurogenic ability. Integrating their dataset to developmental datasets of the zebrafish brain, authors find that radial glia diversity might arise in the embryonic development, which was supported by other findings in the literature. To test the regional origins of the radial glia, authors used parallel gene editing with single-guide RNAs. This approach is a clever way to generate "genetic scars" that can be later identified as lineage barcodes. This strategy suggested that glia might be regionalised within their own niche due to extrinsic factors of that particular brain region. This hypothesis is also supported by velocity studies where authors propose that global glial states could be linked to a neurogenic ability while regional states could be defining structural/other physiological functions. An interesting finding is that snap25+ glia could transdifferentiate into neurons without proliferation while her4+ glia is a proliferative stem cell indicator. To determine the directionality of the cellular physiologies, authors present for the first time scSLAM technique implemented to a live vertebrate brain. Authors state the limitations of the study as dissection of the brain to cause a large spatial region that reduces the cellular resolution mapping to brain microregions, and lineage analyses being limited to the early development.

The study is an important contribution to our understanding of the diversity and organization of neural stem cells in the adult zebrafish brain, which has a high regenerative capacity. The single-cell transcriptomics, lineage tracing, and RNA metabolic labeling techniques to analyze the regulation of neural stem cells in different brain regions over time are valuable technical demonstrations for the field. The documentation of a wide range of neural stem cell states, some of which were specific to certain brain regions while others were found throughout the brain, is important for regenerative medicine. The proposals that the global stem cell states were associated with neurogenic differentiation, with different states being involved in either proliferative or non-proliferative differentiation can be important for clinical ramifications for developmental disorders. The findings have the potential to shed light on the principles of adult stem cell organization and to provide insights into manipulating specific subtypes of neural stem cells for functional purposes.

The writing language in the manuscript appears to be clear and concise. The authors use scientific terminology and technical language appropriate for a research paper. The text is generally well-structured, making it easy to follow the flow of ideas and understand the research methods and results.

In overall, the manuscript is suitable for publication pending on addressing the major comments below.

We thank the reviewer for the excellent summary and the positive feedback on our work.

There are a few inconsistencies or unclear statements in the text that could be improved:

1) In the Abstract, it states, "we combined single-cell transcriptomics of dissected brain regions with massively parallel lineage tracing and in vivo RNA metabolic labeling to analyze regulation of neural stem cells in space and time." However, in the Introduction, it mentions the use of single-cell RNA-seq and lineage tracing, but does not specifically mention in vivo RNA metabolic labeling. The specific details of the methods used may not be fully consistent between the Abstract and Introduction.

We thank the reviewer for this comment. We have now changed the Introduction in order to summarize the methods in a consistent manner.

2) In the Results section, it mentions the identification of 23 neuronal subtypes and 18 radial glia subtypes. However, in the Introduction, it states that there were 23 identified neuronal subtypes and 18 subtypes of immune cells, but does not explicitly mention the number of radial glia subtypes. This discrepancy could be clarified.

We now specify consistently across the main text and the figure legends that subclustering led to identification of 23 neuronal subtypes, 18 radial glia subtypes and 14 subtypes of immune cells.

3) The term "global radial glia" is used to describe certain radial glia states found across all four brain regions. However, in the context of lineage tracing and developmental origin, it is mentioned that there is no early developmental lineage split between global and regional radial glia. This seems contradictory, as the term "global" suggests a distinction in lineage or origin. The concept of "global" radial glia states could be better defined or explained to avoid confusion.

We thank the reviewer for this important comment. We now define the term "global radial glia" explicitly in the Results: These are transcriptional states that we detect across all brain regions, while regional radial glia are transcriptional states that are highly enriched in one of the brain regions.

Hence, our designation of "global" radial glia refers exclusively to the presence of these clusters across different brain regions. Importantly, this does not preclude that these cell states may display a spatially restricted localization within an anatomical region, and this does not necessarily imply a difference in lineage origin. To avoid any confusion, we now explicitly summarize in the Discussion that our combined analysis supports a scenario where radial glia in different brain regions are derived from different clones and have regional identities, but can transition into global states that are linked to neurogenic differentiation.

Overall, while these inconsistencies do not significantly affect the main findings of the study, clarifying and providing more specific information in these areas would enhance the coherence of the text.

We thank the reviewer for these comments that have helped us increase the coherence and clarity of the text.

4) The authors may try to link the identified cellular markers to more referencing and literature as there are many publications that document developmental, homeostatic and pathology-induced radial glial gene expression profiles. Could it be possible to overlap the lineage transitions or trajectory node markers to external references (e.g. gene X in transition from RG subtype A to B in comparison to a disease model or a specific perturbation)? The nature of the advance in this manuscript as is seems to be technical and sample power-related rather than conceptual. Additionally, the advance compared to previous knowledge in the zebrafish field can be better delineated with more rigorous referencing.

We agree with the reviewer that it will be important to link the identified cellular markers more directly to disease models and activity of specific pathways. To systematically address the reviewer's question, we now analyzed the cell state specific expression of all genes for which a neuronal disease phenotype was reported in mice in the Mouse Genome Informatics database (<https://www.informatics.jax.org/>) of the Jackson Laboratory. We chose the mouse due to its more common use as a model organism for disease models.

We selected genes used in mouse models for the following 15 mammalian phenotype ontology annotations related to astrocytes (mammalian homologues of adult zebrafish radial glia), neural stem cells and radial glia: decreased astrocyte number, astrocytosis, increased astrocyte size, radial glia endfoot detachment, decreased radial glia cell number, increased radial glial cell number, ectopic Bergmann glia cells, abnormal neuronal stem cell morphology, impaired neuron differentiation, premature neuronal precursor differentiation, abnormal embryonic neuroepithelial cell proliferation, abnormal astrocyte physiology, abnormal neuronal stem cell physiology, enhanced central nervous system regeneration and impaired central nervous system regeneration. This resulted in a list of 173 unique mouse genes (Dataset EV5), which were translated to zebrafish orthologues with the orthology search function of g:Profiler (<https://biit.cs.ut.ee/gprofiler/orth>). Since not all the obtained genes were expressed in the dataset highly enough to notice meaningful patterns, they were filtered based on average expression in radial glia cells for genes with average expression of 1 in at least one radial glia subtype, resulting in a final list of 54 genes. The expression of these 54 genes across all radial glia subtypes is shown in Fig. R1-1 (included as Fig. S2 in the revised manuscript).

We found that the 54 highly expressed genes are present across all radial glia subtypes at different frequency and expression levels. It is notable, although maybe not surprising, that a particularly high number of these genes are expressed in radial glia subtypes that are involved in proliferation and differentiation, such as both subtypes of proliferating cells as well as *her4++* radial glia. Many of the detected genes in these subtypes were previously known as important for physiology and differentiation of radial glia such as *her4* or the neural differentiation factor *ascl1a*. Particularly highly expressed in the proliferating cells *mki67+* is the *cfl1* gene, which is a depolymerizing factor for actin and whose mouse ortholog was shown as crucial for migration of radial glia and establishment of cortical layers (PMID: 17875668). Structural genes of the cytoskeleton appear as important in other cell types as well, such as the broadly expressed *actb1* and *actb2* as well as the canonical radial glia marker *gfap* (in our dataset only highly expressed in *enkur+* radial glia) and *vim* (neuroepithelial cells). Among the remaining genes, it is interesting to highlight the members of the inhibitor of DNA binding family that inhibit binding of transcription factors (*id1*, *id2a* and *id3*), whose mouse orthologs are important for anchorage of stem cells in their specific niche (PMID: 22522171). Interestingly, the different variants are expressed in different radial glia subtypes, which is in line with the mouse data where, for example, *id2* has been specifically identified as important for development of dopaminergic neurons for olfactory neurogenesis (PMID: 19109490). Another specifically expressed gene is *nrg1* (neuregulin 1), a marker gene for *nrg1+* radial glia in our dataset, which has been shown to have a crucial role in conversion of radial glia to astrocytes in mouse (PMID: 12649319).

We included this discussion in the main text, and we describe details on gene selection and filtering in the Methods.

Figure S2

Fig. R1-1. Comparison to mouse disease models. Expression of genes that are implicated in specific diseases or abnormalities of the nervous system (obtained from the Mouse Genome Informatics database) in radial glia subtypes.

5) Can authors include GO term and KEGG pathways analyses to the manuscript to better define the subtypes for their molecular programs? As is, the manuscript is depicting only gene expressions that may not be fully explaining the physiological relevance of these cell subtypes. Additionally, rigorous discussion on the identified GO terms would enhance the

quality of the paper. There are several interesting hypothesis in this paper but they should be supported by secondary validations (pathway analyses etc). Authors claim the presence of regional specificities of the RGs, but they do not provide any immunostainings or other type of validation data to show that regional cells with specific markers do exist in those regions.

We thank the reviewer for these suggestions. As described below in more detail, we now performed GO, KEGG as well as Reactome pathway analysis of the identified marker genes for all radial glia cell states, and we did fluorescent in situ hybridization experiments to validate the regional specificity of selected subtypes.

For pathway analysis, we entered all cell state marker genes with a p-value of 0.01 or smaller into the analysis. From the resulting pathways, we only considered those that appeared in only one or two radial glia subclusters, in order to focus on molecular programs specific to the subtypes instead of general radial glia terms. We found that the detected cell state enriched pathways were in many cases directly related to the marker genes we had selected to assign a name to the cell state. For instance, *in her++* radial glia we found GO terms related to Notch signaling (in addition to regeneration and development related terms). Similarly, the *mki67+* proliferating radial glia showed enrichment of GO terms related to cell cycle, cell division, DNA replication and chromosome organization. The corresponding KEGG pathways as well as Reactome categories were also significantly enriched. Analogous results could be found for *mcm6+* cells, the second population of proliferating cells we identified. Furthermore, as expected *stra6+* radial glia show retinoic acid signaling in their enriched GO terms. Finally, the *snap25+* radial glia, which we here interpreted to be able to directly differentiate into neurons, have multiple enriched neuron related GO terms including neurotransmitter transport, microtubule organization, voltage-gated sodium channels and other categories related to the formation of synapses. We included the full results of this analysis in Dataset EV4.

To validate our approach for identification of region-specific radial glia subtypes based on dissection of brain regions and scRNA-seq, we now also performed RNAscope experiments for selected marker genes. This analysis, which is included in the revised manuscript in Fig. 2C and in Fig. S4, is shown below as Fig. R1-2. The in situ hybridization experiments are in full agreement with our sequencing data, thereby validating the approach. Specifically, we found that, as expected, expression of *apof* was restricted to the mesencephalon, *prss35* was only detected in the rhombencephalon, *AL954697.1* was specific to the diencephalon, and *crabp1b* was only detected in the rhombencephalon.

Fig. R1-2. RNAscope for selected marker genes in medial brain sections. A. Expression of marker genes in radial glia subtypes, based on scRNA-seq. B-D. Expression of *apof*, *prss35*, AL954697.1 and *crabp1b* is restricted to the expected brain regions.

6) The authors use *gfap*:GFP transgenic animal that marks the astroglia-like cells in the zebrafish brain. This line does not mark all types of radial glia. Can a quality control step be implemented showing the colocalization of *gfap*:GFP cells with other glial markers to have a quality measure?

We thank the reviewer for this important comment. We agree that the *gfap*:GFP transgenic line does not mark all types of radial glia with the same efficiency. However, this was not a major concern in our study design, since the sorted cells are only a minor contribution to the dataset and also only account for a minor fraction of the radial glia. Furthermore, we used rather wide sorting gates in FACS in order to minimize selection biases. As expected, and as shown in Fig. R1-3, the contribution of sorted *gfap*:GFP cells varies between the different cell states, but due to the relatively small amount of sorted cells this has only a minor influence on the overall abundances of the radial glia states.

Fig. R1-3. Fraction of sorted cells in the different radial glia subtypes.

Please also note that we exchanged Fig. S3B in the revised manuscript, since the previous version, while technically correct, could be misinterpreted to suggest a larger contribution of sorted cells to the radial glia pool than we actually had.

7) The power of the studies are fantastic. However, the methodological details can be explained more and documentation of the sample characteristics can be significantly improved. The corresponding author is a leading expert on the topic, but the reviewer GEO link was not provided. The reviewers cannot comment on the data content and quality.

We thank the reviewer for the positive feedback. We now included a link to the GEO submission and to the github repository, and we expanded the Methods section of the manuscript as well as the information on sample characteristics. Specifically, we added more information about sample quality in Dataset EV1, and we provide the complete list of marker genes used for radial glia subtype identification in Dataset EV3. Furthermore, we expanded the sections about data pre-processing and clustering in the Methods.

Reviewer #2:

Comments on Paper Mitic et al., Dissecting the spatiotemporal diversity of adult neural stem cells.

General comments

The authors based all of their conclusions on sequencing data to which they:

a. do not show any plots to indicate the resolution and the quality of sequencing (example of such plots would be ones showing mitochondrial genes distribution).

We thank the reviewer for this suggestion. We now included information about the quality of the sequencing data in Dataset EV1.

b. do not indicate how many replicates are used, does each biological replicate consist of only one zebrafish brain or are they pooled (and how many are pooled)

We now included this information in Dataset EV1.

c. show no validation at all of any sequencing data on the clustering of different regions. Such validations are essential and could have been easily performed with immunostainings or RNAscope for example.

We thank the reviewer for this important comment. To validate our approach for identification of region-specific radial glia subtypes based on dissection of brain regions and scRNA-seq, we now performed RNAscope experiments for selected marker genes. This analysis, which is included in the revised manuscript in Fig. 2c and S4, is shown below as Fig. R2-1. The in situ hybridization experiments are in full agreement with our sequencing data, thereby validating the approach. Specifically, we found that, as expected, expression of *apof* was restricted to the mesencephalon, *prss35* was only detected in the rhombencephalon, *AL954697.1* was specific to the diencephalon, and *crabp1b* was only detected in the rhombencephalon.

Fig. R2-1. RNAscope for selected marker genes in medial brain sections. A. Expression of marker genes in radial glia subtypes, based on scRNA-seq. B-D. Expression of *apof*, *prss35*, *AL954697.1* and *crabp1b* is restricted to the expected brain regions.

Specific Figure comments

Fig.1

It is not clear to the reviewer how the clustering was performed. A more detailed description on how the clustering was done is needed. Just having 1-3 genes is not strong enough to claim that these are distinct types (eg. in several GABAergic clusters, the difference is based on only one marker expression, which could be stochastic expression.)

We agree with the reviewer that 1-3 genes would not be sufficient to claim distinct cell states. In the previous version of the manuscript we had only shown a small number of selected marker genes (Dataset EV2), which may have been misleading. In the revised manuscript we now also included the complete list of marker genes per radial glia subtype (Dataset EV3), which shows that each cell state is characterized by a considerable number of differentially expressed marker genes. Furthermore, in the Methods we expanded the description of how the clustering was performed.

In order to further strengthen the clustering analysis and its interpretation, we now also performed GO, KEGG as well as Reactome pathway analysis of the identified marker genes for all radial glia cell states. We entered all cell state marker genes with a p-value of 0.01 or smaller into the analysis. From the resulting pathways, we only considered those that appeared in only one or two radial glia subclusters, in order to focus on molecular programs specific to the subtypes instead of general radial glia terms. We found that the detected cell state enriched pathways were in many cases directly related to the marker genes we had selected to assign a name to the cell state. For instance, in *her4++* radial glia we found GO terms related to Notch signaling (in addition to regeneration and development related terms). Similarly, the *mki67+* proliferating radial glia showed enrichment of GO terms related to cell cycle, cell division, DNA replication and chromosome organization. The corresponding KEGG pathways as well as Reactome categories were also significantly enriched. Analogous results could be found for *mcm6+* cells, the second population of proliferating cells we identified. Furthermore, as expected *stra6+* radial glia show retinoic acid signaling in their enriched GO terms. Finally, the *snap25+* radial glia, which we here interpreted to be able to directly differentiate into neurons, have multiple enriched neuron related GO terms including neurotransmitter transport, microtubule organization, voltage-gated sodium channels and other categories related to the formation of synapses. We included the full results of this analysis in Dataset EV4.

There is inconsistency in switching between the terms cell types, cell clusters and cell state throughout the text, which makes it difficult to understand and follow.

We thank the reviewer for this important comment. We use the term "cell type" to refer to major categories of cells (neurons, radial glia, immune cells, oligodendroglia), and we use the term "cell state" (or "subtype") for subclusters (e.g. specific subtypes of radial glia). We use the term "cell cluster" to refer to the outcome of a computational clustering analysis, independent of our biological interpretation as a cell type or cell state. We now define these terms at the beginning of the Results section, and we improved our consistency in using these terms throughout the manuscript.

Fig.2

It is not clear if all sorted cells were integrated or only radial glial cells.

We now specify in the legend of Fig. S3 that all sorted cells were integrated, including sorted non-radial glia cells. However, only cells from dissected brains (i.e. excluding undissected whole brains and sorted cells) were used to analyze spatial location in Fig. 2A.

The authors mention using a threshold of 70% without any explanation on how this threshold was determined.

We now included the following explanation in the text: "We set the cutoff to 70% to account for the limited accuracy of dissection, as evidenced by our data: For example, the known cerebellar population of Bergmann glia (cluster 6) is accurately found in the rhombencephalon but only at ~80%, suggesting that a certain degree of cross-region contamination does occur and the cutoff needs to be permissive."

However, we wish to note that choosing a cutoff is somewhat subjective. We therefore also show a non-binarized version of the regional specificity of radial glia states, where we plot the telencephalon-specific fraction of neuronal and radial glia subtypes on a continuous scale (Fig. 4a).

Fig.3

The reviewer found this figure very hard to follow and understand

It is not clear how one clone can be considered regional and global at the same time!

We apologize for the unclear explanation. We have now improved the figure and expanded our explanation in the text in order to address the reviewer's comment:

1) We now define the term "global radial glia" explicitly in the Results: These are transcriptional states that we detect across all brain regions, while regional radial glia are transcriptional states that are highly enriched in one of the brain regions. Hence our designation of global and regional radial glia refers exclusively to the presence of these cell states across different brain regions. Importantly, this does not necessarily imply a difference in lineage origin. In fact, the clones detected by our lineage analysis consist of regional as well as global radial glia. This shows that regional and global radial glia do not originate from separate lineages, and this finding is in agreement with our later observation (in Fig. 4 and 5) that regional radial glia can transition into global states.

To avoid any confusion, we now explicitly summarize in the Discussion that our combined analysis supports a scenario where radial glia in different brain regions are derived from different clones and have regional identities, but can transition into global states that are linked to neurogenic differentiation.

2) Addressing what we believe was the main source of confusion, we now specify that our lineage analysis was performed with undissected whole brains and therefore does not contain direct spatial information: While we can infer the spatial location of cells in regional states, we do not have information about the spatial location of cells in global states. The interpretation of the spatial structure of clones is hence purely based on regional states. Taking this into account, we found that most lineage clones displayed a clear spatial structure, with clones being dominated by regional cells from one or at most two brain regions.

In summary, the majority of clones have a regional identity (i.e. regional subtypes are mostly from one brain region), but within each clone we also find global states. We strongly suspect that, if the regional cells in a specific clone are all from e.g. the telencephalon, the global cell states will also be located in the telencephalon, but we do not have a direct experimental measurement for this.

There is no sequencing data of the clones recovered shown. How do the authors control for the mutations they get with the scarring method? Relying only on the lethality of embryos does not make sense at all, as some baseline lethality is expected anyway when injecting anything into zebrafish embryos at 1-cell stage.

In the revised manuscript we now included alignments of observed CRISPR scar sequences in Fig. S5D in order to establish a more direct link between the sequencing data and the clonal analysis. This plot is shown below in Fig. R2-2.

Fig. R2-2. Alignments of selected CRISPR scar sequences to the wildtype target sites. The CRISPR/Cas9 target sites are located in the 3' UTR of endogenous genes, as shown for *rp139* on the left. Each clone ID is defined by two 40 base-pair sequence IDs, corresponding to the two alleles of the gene. (included in the revised manuscript as Fig. S5D).

We are not sure we fully understand the reviewer's question about the control of mutations. The mutations are localized to the specific target regions (sites in the dTomato transgene and in the 3' UTRs of selected housekeeping genes). We rely on non-homologous end joining to generate a diversity of sequences that can be used as lineage barcodes. Targeting 3' UTRs of endogenous genes is potentially risky, but we did not detect any changes in lethality (compared to the normal baseline lethality), and we did not observe any phenotypes. In the revised manuscript, we describe the lack of changes in lethality and of observable phenotypes in more detail.

Fig.4

The reviewer found it hard to 'clearly distinguish' the trajectories in Fig 4A as the authors describe in the text.

We apologize for the lack of clarity in our graphical representation of the results. We have now improved the layout of Fig. 4A to indicate the two trajectories more clearly: In the UMAP layout we observe two separate connections between radial glia and neurons – one at the top and the other one at the bottom of Fig. 4A. The directionality, and hence the interpretation as differentiation trajectories, is supported by RNA velocity arrows, and the two trajectories involve different radial glia states (*snap25+* radial glia for the upper trajectory, *her4++* radial glia for the lower trajectory). The new layout is shown below in Fig. R2-3.

Fig. R2-3. Direct and proliferative differentiation follow separate trajectories and involve distinct radial glia states (Fig. 4A). We now highlight the two separate transition regions in zoom-ins.

Error bars are not clear at all and seem to be shifted/moved

We have fixed this.

The authors indicate in the text that intron/exon based RNA velocity has its caveats and they even cite several papers discussing these caveats at different steps while performing RNA velocity. To remedy this, the authors employ metabolic labelling of nascent RNAs. However the papers cited by the authors point to much deeper problems in RNA velocity pipelines than just the use of introns and exons. For example the way that data is processed by RNA velocity methods is a large problem. As a consequence, Gorin et al. advise against normalisation, imputation and most things that make up RNA velocity pipelines. As a consequence, I think that it is unlikely that using SLAMseq will remedy the problems in how RNA velocity data is analysed. Moreover, how the authors perform RNA velocity analysis is not indicated at all.

As the reviewer correctly notes, there are multiple caveats in RNA velocity analysis, and not all of these issues can be solved by using RNA metabolic labeling instead of intron/exon ratios. In the revised manuscript we now discuss the advantages of RNA velocity based on metabolic labeling as well as the remaining unsolved computational challenges in a more nuanced way by including the following paragraph:

"This approach has at least two advantages compared to analysis of intron/exon ratios: 1) The computation is based on all detected genes, not only on those with introns that are detectable by scRNA-seq. 2) The time window of labeling is well defined and is the same for all genes, in contrast to gene-dependent splicing rates. However, other potential computational issues related to model biophysics, fitting, normalization and scaling remain to be addressed (refs)."

We here used an existing package ("Dynamo", Qiu et al., Cell, 2022) to compute RNA velocity based on scSLAM-seq. In the revised manuscript we briefly describe the computational approach in the main text, in addition to a more detailed protocol in the Methods. While we are convinced there is room for further computational development in RNA velocity based on scSLAM-seq, we also believe it would be out of scope for this manuscript to develop a new mathematical and computational framework. In fact, we have started a collaboration with the computational group of Laleh Haghverdi to further improve

RNA velocity calculations in the future. Importantly, even though existing computational approaches for calculating intron/exon based as well as metabolic labeling based RNA velocity are not perfect, it is reassuring that both approaches support the directionality of the two differentiation trajectories.

Fig.5

Plots in this figure are not explained in the text

Some validation that the notch inhibition treatment is actually working is lacking.

We now expanded our explanation of the plots. Furthermore, we show in Fig. S7A of the revised manuscript that the Notch inhibition treatment is indeed working by analyzing the differential expression of Notch target genes (plot shown below in Fig. R2-4).

Notch downstream genes expression - pseudobulk

Fig. R2-4. Notch inhibition quality control (Fig. S7A). Heat map of average expression of Notch pathway genes in all radial glia clusters (based on log-normalized expression data) in the control (DMSO) and Notch inhibition (DAPT) datasets.

Reviewer #3:

The manuscript entitled "Dissecting the spatiotemporal diversity of adult neural stem cells" by Mitic et al. describes the diversity of radial glia cell states in the adult zebrafish brain. This work is mainly accomplished by use of single-cell RNA sequencing where the authors profiled cells from microdissections of four major brain regions (telencephalon, diencephalon, mesencephalon, and rhombencephalon). The authors present a substantial dataset of 100,000+ cells from zebrafish brains from which they describe several radial glia (RGs) sub-clusters. They classify RGs based on their regional distribution and gene expression profiles. These analyses revealed two spatial distributions of RGs, those that are mostly confined to one region ("regional") and those that are found in multiple regions ("global"). Additionally, the authors reveal several sub-clusters of RGs based on gene expression, for example proliferating RGs, *snap25* expressing RGs, and *her4* expressing RGs. Using an enhanced version of their previously published method for lineage tracing in zebrafish, the authors show that regional and global RGs are not from different lineages and instead clonal families contain both global and regional RGs. Additionally, the authors use SLAM, scSLAMseq, and Notch inhibition experiments to find that different gene expression based clusters of RGs have different transition probabilities to other cell types/states.

While I believe the data presented and analysis thereof will be beneficial to the community I do have major concerns. One general concern is that the data analysis and data presentation appear preliminary and partially confusing. More work needs to go into extracting information from the data and presenting it in a more clear and concise way. A further major concern is the lack of validation (spatial analysis, staining) of cell populations identified using single-cell transcriptomics. Below I am listing my questions and concerns in more detail:

We thank the reviewer for the positive feedback and for the constructive criticism. As discussed in detail below, we now improved data analysis and presentation, and we added extensive validation experiments using fluorescent in situ hybridization.

1) The authors present an impressive amount of single-cell data across the zebrafish brain; however, it would be beneficial for the authors to include some spatial information on how different clusters across the different brain regions are arranged within the tissue. Spatial information would be particularly of interest for the sub-clusters of radial glia cells, including the small group of neuroepithelial cells or the *enkur+* cluster which is speculated to be lining the ventricle (do you find this cluster in the ventricles of other regions outside the telencephalon?). How do subclusters of radial glia relate in space to each other even within a given region?

We thank the reviewer for this important comment. To validate our approach for identification of region-specific radial glia subtypes based on dissection of brain regions and scRNA-seq, we now performed RNAscope experiments for selected marker genes. This analysis, which is included in the revised manuscript in Fig. 2c and in supplemental figure S4, is shown below as Fig. R3-1. The in situ hybridization experiments are in full agreement with our sequencing data, thereby validating the approach. Specifically, we found that, as expected, expression of *apof* was restricted to the mesencephalon, *prss35* was only detected in the rhombencephalon, *AL954697.1* was specific to the diencephalon, and *crabp1b* was only detected in the rhombencephalon.

Furthermore, we also used RNAscope to characterize the expression patterns of *enkur+* radial glia and neuroepithelial cells. The expression of the marker genes *enkur* and *clu* in sections taken from the areas lining the ventricle is shown in the revised manuscript in Fig. S4E (and below in Fig. R3-1E).

We agree with the reviewer that the microscopy data would allow us to perform a more fine-grained analysis of spatial relationships also within a given region. However, we believe that a more detailed analysis of the spatial relationship of the radial glia states within individual brain regions would be out of scope for our manuscript, which is focused on understanding the relationship of radial glia states in the framework of differentiation trajectories.

Fig. R3-1. RNAscope for selected marker genes in medial brain sections. A. Expression of marker genes in radial glia subtypes, based on scRNA-seq. B-D. Expression of *apof*, *prss35*, *AL954697.1* and *crabp1b* is restricted to the expected brain regions. E. Expression of *clu* and *enkur* in the areas lining the ventricle in the telencephalon and diencephalon (medio-dorsal wall of the ventricle in the telencephalon and ventral wall in the diencephalon).

I could not find the numbers of cells belonging to each major cell type (neurons, radial glia, immune cells, etc.) or to the subtypes of each cell type. These numbers should be stated either in the text or in the figures (n=xx). Additionally, it would be good to know how many cells are contributed from which experiments (unbiased cell profiling vs FACS enriched experiments, etc.). Further, please add information regarding how many replicates are used for these experiments.

We now included this information in Dataset EV1 (number of animals and replicates, total number of cells per experiment, including the different conditions: whole brain, dissected brain, FACS), in the legend of Fig. S1 (number of cells in major cell types), and in Dataset EV2 (number of cells in radial glia subtypes).

Figure 2 - please add an indicator of which clusters are global or regional. This would make it easier for the reader to understand the different sub-clusters that are highlighted.

We had included an indicator for regional/global clusters in Fig. 2 (in orange and blue below the bar graphs), but we agree that this was not clear enough. We now indicated this more clearly.

I understand the authors use SoupX to remove "contaminating" mRNAs. Do the authors identify why this is necessary or which genes they find problematic? Moreover, the authors identify one sub-cluster of radial glia with snap25+ expression, could this be from high ambient RNA from neurons, RG-neuron doublets or is this true signal from radial glia cells? This cluster is of particular interest in the subsequent analysis using scSLAMseq thus the importance of showcasing the "true" signal (for example with spatial methods) of snap25 would be critical.

Our main reason for using SoupX was that we detected a variable amount of mitochondrial reads between samples, and that we wanted to allow integration of methanol-treated samples. Another concern we had is that there could be a different level of background ambient RNA in FACSed vs non-FACSed cells.

The reviewer raises an important point regarding the existence of snap25+ radial glia. Since the marker gene snap25 is also expressed in neurons, one can speculate that the cluster of snap25+ radial glia might be an artifact caused by ambient RNA or doublets containing a neuron and a radial glia cell. However, there are several lines of reasoning that rule out this possibility: We do not detect snap25 in other radial glia from the same experiments, which argues against ambient RNA. Furthermore, snap25+ radial glia do not appear to be doublets, since they have normal UMI counts and are not flagged by doublet detection algorithms. Additionally, snap25+ radial glia are also characterized by expression of additional marker genes like *ggctb* with different expression pattern across neuronal subtypes (Fig. R3-2), which again argues against doublets. Most importantly, our new RNAscope data directly confirms the existence of snap25+ radial glia, since we find co-expression of *snap25* and *ggctb* in a subset of radial glia (Fig. R3-3, included in Fig. S4 in the revised manuscript).

Fig R3-2. Expression of markers genes *fabp7a*, *gfap* *snap25a* and *ggctb* across radial glia and neuronal sub clusters. A) radial glia subclusters, B) neuronal subclusters

Fig. R3-3. Validation of *snap25*⁺ radial glia by RNAscope in medial brain sections. A. Expression of *snap25*⁺ radial glia marker genes *snap25* and *ggctb* in radial glia and neuronal subtypes, based on scRNA-seq. B. As expected, we find co-expression of *snap25* and *ggctb* in a subset of radial glia. C. Radial glia expressing *snap25* can be detected also without the additional marker *ggctb*.

Could the authors explain why the diencephalon was the least accurate of the dissections and possibly add this rationale to the sentence they already present in their text or to the methods?

We thank the reviewer for this important question. The diencephalon has three borders to the neighboring brain regions (see cartoon in Fig. 2A), and none of these borders is very clear. By contrast, the borders between telencephalon, mesencephalon and rhombencephalon are much clearer. We now included this statement in the Methods.

Figure 2A is hard to follow, for example the authors highlight Bergmann glia (cluster 6) and while it is described in the text as Bergmann glia the figure still calls it RG prss35⁺. The

authors should make these annotations more consistent and the figure easier for the reader to navigate.

We thank the reviewer for this comment. We now made sure to use consistent names for the subtypes, and we improved the layout of Fig. 2A by indicating the assignment to global versus regional states more clearly.

Figure 3A - authors should avoid pie charts; it is hard to understand the radial glia sub-cluster or the integrated data cluster composition across the different timepoints which makes it hard to interpret the authors findings.

We now changed this plot (as well as the pie charts in Fig. 4) to bar graphs. We furthermore improved data analysis and visualization in Fig. 3A, as discussed in response to the reviewer's following comment below.

The authors state that radial glia diversity is already established at embryonic and early larval stages, but the UMAP in Figure 3A appears to have many cells that appear primarily from the adult. This part of the analysis could use some more exploration. For instance, an additional supplemental figure that shows why earlier larval timepoints were excluded or if there are radial glia cell clusters from the integrated data that are primarily from adult. Are there any notable differentially expressed genes between adult and larval stages in clusters that are more equal in their cluster contribution (example: RG *foxn4+* or the NE cluster)

We now expanded the comparison of embryonic/larval and adult radial glia in several ways and updated Fig. 3A in the revised manuscript (shown below in Fig. R3-4):

- We now also included earlier embryonic time points (20, 24 and 36 hours post fertilization). We still excluded the earliest time points 12, 14, 16 and 18 hpf, since no radial glia were annotated at these stages in the original publication (Raj et al.). Inclusion of the time points 20, 24 and 36 hpf did not change our conclusions regarding the early establishment of radial glia diversity.
- We now show in Fig. 3A that the main difference between embryonic/larval and adult stages is a strong enrichment of proliferating radial glia at developmental stages. However, the embryonic/larval samples also contribute to all adult radial glia clusters.
- We now switched from pie charts to bar graphs for better readability of the changes in cell state abundance, and we included adult radial glia in the comparison. Of note, the radial glia state related to direct differentiation (*snap25+*) is enriched strongly in the adult brain, in contrast to the radial glia state related to proliferative differentiation (*her4++*).

Fig. R3-4. Developmental emergence of radial glia diversity (Fig. 3A). Data integration of adult radial glia with developmental radial glia and neural progenitors from Raj et al, 2020.

The reviewer raises the intriguing question whether there are notable differentially expressed genes between adult and embryonic/larval stages within individual clusters. While this is an interesting question, we believe the data is not ideally suited for such a fine-grained analysis, since we would compare datasets prepared in different labs with different dissociation protocols, which could lead to batch effects that would be difficult to detect. We would therefore prefer to forgo such an analysis.

Figure 3C - this is hard to interpret; labeling the clones (1, 2, 3, ...) could add the reader. Regarding the clone bar annotation, what does it mean when cells are not in a clone bar? Are these single cells with no family? If this is true could this figure be in the supplement and instead highlight the clonal families in the main figure (highlight the sizes of clonal families and the number of cells belonging to different annotation groups like cell type and region)? Can the color choices for Figure 3 and Figure S3C be adjusted to aid the reader with the color bars (particularly for the cell type annotations).

We now made the following graphical changes in order to help readers interpret the figure: In Fig. 3C (shown below in Fig. R3-5) we labeled the clones with numbers, we changed the colors (in particular for cell type annotation), we show selected clones in close-up, and in Fig. S5C (previously S3C) we now included actual sequence information in order to link the analysis more directly to the raw data.

Fig. R3-5. CRISPR lineage tracing (Fig. 3C). The depicted heatmap shows hierarchical clustering of radial glia and neurons based on identified lineage scars for 331 cells from a single brain.

As the reviewer correctly noted, there is a small number of cells that is not assigned to a clone. There are two main reasons for this: 1) Due to the inherent sparsity of scRNA-seq we do not necessarily detect all lineage scars. This leads to a small fraction of false negatives. 2) In the annotation we did not indicate very small clones (e.g. the one between clone 4 and 5 for better readability).

Figure 4B- The pie charts are difficult to understand, the percentages for the dominant cell type that a cell transitions to could be labeled as well as a legend for the cell type annotations.

We now changed the graphical representation to bar graphs, which we think makes it easier to see the percentages of inferred cell state transitions.

Figure 4C - RG *her4++* appears to transition to the proliferating cells but is also a high amount of cells to neurons_5. This probability actually appears higher than the *snap25+* RGs that transition to neurons (which have probabilities to other *snap25+* and *id2b+* RG clusters). Generally, the plots in 4B and 4C are difficult to understand the author's analysis and conclusions.

We now improved the layout of Fig. 4B,C for easier interpretation (shown below in Fig. R3-6). However, the two analyses in Fig. 4B and 4C indeed reveal several transitions in addition to "*snap25+* RG  neurons" and "*her4++* RG  proliferating cells  neurons". We believe this is unavoidable and to be expected for two reasons: First, the transitions are not directly measured but inferred based on transcriptomic similarity and RNA velocity, so a certain level of background signal is to be expected. Second, and more importantly, we here analyze steady-state conditions in the unperturbed adult brain, so there is only a limited amount of ongoing neurogenesis. It is therefore not surprising that, in addition to neurogenic differentiation, we can also detect other transitions, including "backwards" from global to regional radial glia states.

We now discuss this in the main text, and we explain more clearly that our main finding in this analysis is that there are transitions from *her4++* radial glia to proliferating cells (and further on to neurons), while no transitions of *snap25+* radial glia to proliferating cells can be detected (only direct transitions to neurons).

As the reviewer correctly notes, we detected a large fraction of transitions from *her4++* radial glia to neurons (higher than the fraction of transitions from *her4++* radial glia to proliferating cells) in the CellRank analysis in Fig. 4C (R3-6 B). This is a result of the CellRank algorithm, which attempts to reconstruct full trajectories, potentially across multiple cell states, and which here identifies neurons as the final stage. This is in contrast to the analysis in Fig. 4B (R3-6 A) where we analyzed only one-step transitions, and where, in agreement with our interpretation, a much lower degree of transitions from *her4++* radial glia to neurons is detected. However, we agree with the reviewer that this was not explained well in the manuscript, and we now improved our discussion of this topic in the main text.

A transition probability from each cluster

B CellRank absorption probabilities for each cluster

Fig. R3-6. Transition probabilities (Fig. 4B,C). A. Transition probabilities (based on RNA velocity) between highlighted cell states. For each cell within the cell state, the maximum transition probability and the corresponding cell state to which that cell is most likely to transition were determined. B. Prediction of absorption probability for each of the highlighted cell state groups based on the CellRank algorithm.

Figure 5 figure legend - The reference Fig S5C, however this figure does not exist.

We apologize for this mistake. We have corrected this reference (Fig. S5 from the previous version is now Fig. S7).

Figure 5D - this panel is hard to interpret. The variation in the fraction of snap25+ RGs involved in in direction differentiation is not clear from the two UMAPs of the treatment and control.

We agree with the reviewer that the effect of Notch inhibition on cell state abundances is unclear. It is well established in the literature that Notch signaling prevents proliferation, and that Notch inhibition in turn leads to an increase in proliferative differentiation. One could therefore expect (and this was indeed our initial motivation for this experiment) that the number of proliferating cells would increase upon Notch inhibition, and that the ratio between proliferative and direct differentiation would change. However, we did not detect a clear difference and only observed a mild increase of *snap25+* radial glia (the statistical significance of which remains unknown), even though the Notch inhibition was clearly effective (Fig. S7). There are several potential reasons for this: 1) It is possible that cells move along the proliferative trajectory at higher speed upon Notch inhibition, without changes in cell abundance. 2) It is possible that at the time point we analyzed (two days after the onset of Notch inhibition), compensatory mechanisms have already set it to

normalize cell state ratios. 3) The number of cells and the number of samples may not be sufficient to detect the effect.

In summary, we had hoped to be able to visualize differences in proliferative versus direct differentiation with this experiment. However, this is not required for supporting the main conclusions of this manuscript (existence of separate differentiation trajectories involving different radial glia states for direct and proliferative differentiation), and we therefore believe it would not be warranted to investigate this further. We now discuss this in more detail in the text.

When inhibiting Notch and thus increasing the amount of direct differentiation, is there an observable difference in the neuronal output sub-clusters, are particular neuronal populations associated with direct neurogenesis?

The reviewer raises an interesting question: Do proliferative and direct differentiation produce the same neuronal subtypes? The most straightforward way to approach this question would be to check if there are differences in neuronal cell type abundance upon Notch inhibition in the scSLAM-seq datasets, as suggested by the reviewer. However, a major limitation of comparing the two conditions is the lack of replicates (n=1 for control and Notch inhibition), so it would not be possible to distinguish batch effects from real biological differences. Furthermore, the number of cells is relatively low in the scSLAM-seq dataset, so the resolution of neuronal subtypes would be limited. Finally, we would expect that any potential effect would probably be small, since neurons are long-lived and the Notch inhibition treatment was short compared to their lifetime.

We therefore reasoned that it might be more promising to detect differences in neuronal output between the proliferative and non-proliferative trajectories in the much bigger scRNA-seq using CellRank analysis. Compared to the analysis in Fig. 4C, we now analyzed the inferred trajectories with higher resolution, taking different neuronal outputs into account. When comparing the inferred transition probabilities between *snap25+* radial glia and proliferating cells we did not find major differences in the preferred neuronal attractor states – states Neurons_5 and Neurons_8 in both cases receive the highest fractions apart from self-transitions. While we cannot rule out that this negative result is due to limitations of the inference algorithm, we can summarize that CellRank did not provide evidence for different neuronal outputs of the proliferative vs direct differentiation trajectory. This analysis is shown in Fig. R3-7 and R3-8.

Fig. R3-7: Transition probabilities of snap25+ radial glia.

Fig. R3-8: Transition probabilities of proliferating cells.

As a final analysis, we now also applied CellRank to the scSLAM-seq datasets. While the cell numbers are much smaller, we reasoned that the RNA labeling data and the ability to compare control and Notch inhibition might still be advantageous for addressing the reviewer's question. For this we leveraged the scSLAM-seq based velocity analysis (Fig. 5) and ran CellRank on the resulting velocities. The results were very similar to the non-SLAM-seq dataset presented above: proliferative neurogenesis as well as direct differentiation seem to lead to similar neuronal states (see Fig. R3-9 and R3-10, please note that this is a different analysis and therefore the state names and numbers do not match with the data shown in R3-7 and R3-8). Keeping the limitation of relatively low cell numbers in mind, the only major observable difference was a reduction of non-self transitions of snap25+ radial glia, suggesting that these cells indeed respond in a compensatory manner upon Notch inhibition. In summary, we did not find evidence for differential neuronal outputs between proliferative and direct differentiation.

Fig. R3-9: Transition probabilities of snap25+ radial glia upon notch inhibition as well as in a non-inhibited control experiment.

Fig. R3-10: Transition probabilities of proliferating cells upon notch inhibition as well as in a non-inhibited control experiment

19th Dec 2023

Manuscript Number: MSB-2023-11757R

Title: Dissecting the spatiotemporal diversity of adult neural stem cells

Dear Philipp,

Thank you for sending us your revised manuscript. We have now heard back from the three reviewers who were asked to evaluate your revised study. As you will see below, the reviewers mention that most of their comments have been satisfactorily addressed. Reviewer #2 lists a few remaining concerns, which we would ask you to address by providing some clarifications and performing some text/figure modifications in a final round of minor revisions. We would also ask you to address some remaining editorial issues listed below.

- Our data editors have noticed that the information related to n is missing in the legend of figure 4c, please correct this.
- The funding information provided in the manuscript text need to match the information entered in the online submission system. Currently the following info missing from the submission system: PhD fellowship from Studienstiftung des deutschen Volkes.
- The keywords need to be reduced to 5.
- Please remove the 'Authors Contributions' from the manuscript. The 'Author Contributions' section is replaced by the CRediT contributor roles taxonomy to specify the contributions of each author in the journal submission system. Please use the free text box in the 'author information' section of the online submission system to provide more detailed descriptions if needed (e.g., 'X provided intracellular Ca⁺⁺ measurements in fig Y').
- Please include page numbers missing in the Appendix Table of Contents. In the Appendix Table of Contents and figure legends, the figures should be labeled as Appendix Figure S1-S7 (currently they are wrongly called Figure S1-S1 or Supplementary Figure S1-S7).
- The Figure Legends should be moved at the end of the text, after the References.

Please resubmit your revised manuscript online, with a covering letter listing amendments and responses to each point raised by the referees. Please resubmit the paper ****within one month**** and ideally as soon as possible. If we do not receive the revised manuscript within this time period, the file might be closed and any subsequent resubmission would be treated as a new manuscript. Please use the Manuscript Number (above) in all correspondence.

Click on the link below to submit your revised paper.

All my best wishes for the upcoming Holidays,

Maria

Maria Polychronidou, PhD
Senior Editor
Molecular Systems Biology

If you do choose to resubmit, please click on the link below to submit the revision online before 18th Jan 2024.

IMPORTANT:

Please note that corresponding authors are required to supply an ORCID ID for their name upon submission of a revised manuscript (EMBO Press signed a joint statement to encourage ORCID adoption).
(<https://www.embopress.org/page/journal/17444292/authorguide#editorialprocess>)

Currently, our records indicate that the ORCID for your account is 0000-0002-2826-8290.

Link Not Available

*** PLEASE NOTE *** As part of the EMBO Press transparent editorial process initiative (see our Editorial at <https://dx.doi.org/10.1038/msb.2010.72> , Molecular Systems Biology will publish online a Review Process File to accompany accepted manuscripts. When preparing your letter of response, please be aware that in the event of acceptance, your cover letter/point-by-point document will be included as part of this File, which will be available to the scientific community. More information about this initiative is available in our Instructions to Authors. If you have any questions about this initiative, please contact the editorial office (msb@embo.org).

Reviewer #1:

Authors addressed all my comments satisfactorily. I see the manuscript suitable for publication.

Reviewer #2:

The authors have partially addressed previous concerns. Specifically:

Sequencing quality control and replicates are shown in a supplementary table and the figure legend of figure 1, however, it is not indicated which animals were used for which experiment and how many replicates were used per experiment or if samples were simply pooled. All this information should be transparently shown in each figure legend stating which specific subset of the overall dataset is used.

Specific Figure comments

Fig.1

The information given in the figure legend and EV1 show that samples of different origin and pooling strategy are used to create the UMAP. To show a reasonable clustering independent of sample handling and pooling the different replicates should be shown in a separate UMAP as supplementary information in order to exclude any batch effects on the clustering.

Fig.2

indication of replicates/pooled samples used are missing in the figure legend. This should be added.
The RNAscope data should be quantified in addition to showing representative images.

Fig.4

The concerns raised regarding RNA velocity: the authors write in their response that they are aware of the computational issues of RNA velocity and that they are working on improving them in the future but outside the scope of this manuscript. I can understand the argumentation of the authors however, have trouble seeing the value of a figure based solely on the trust on RNA velocity. Besides the concern regarding distinguishing the trajectories was not with regard to the UMAP representation, but rather to the trajectories shown, which are based on the authors interpretation and the line could as "clearly" also be drawn in different positions on the UMAP. Given that scSLAM-Seq is used to validate the results in Fig. 5 in this UMAP, the trajectories described in Fig.4 seem even less clear.

Fig.5

According to EV1 only 1 replicate of pooled brains was used per condition. This should clearly be stated in the figure legend

Reviewer #3:

I appreciate the work that the authors have put into revising the manuscript such as the addition of validation data using RNAscope. It is a nice piece of work and I support its publication. Some of my concerns still remain, but the authors might be right in their answer that this is beyond the scope of this work.

Reviewer comments

Reviewer #1:

Authors addressed all my comments satisfactorily. I see the manuscript suitable for publication.

We thank the reviewer for the positive feedback and for the constructive comments in the first round of revision.

Reviewer #2:

The authors have partially addressed previous concerns. Specifically: Sequencing quality control and replicates are shown in a supplementary table and the figure legend of figure 1, however, it is not indicated which animals were used for which experiment and how many replicates were used per experiment or if samples were simply pooled. All this information should be transparently shown in each figure legend stating which specific subset of the overall dataset is used.

We now provide this information in the figure legends. Specifically, we indicate the sample IDs of the libraries that were used in the figures, and Dataset EV1 contains further information about the individual libraries (including the number of animals which were pooled).

Specific Figure comments

Fig.1

The information given in the figure legend and EV1 show that samples of different origin and pooling strategy are used to create the UMAP. To show a reasonable clustering independent of sample handling and pooling the different replicates should be shown in a separate UMAP as supplementary information in order to exclude any batch effects on the clustering.

We agree with the reviewer that it is important to rule out that the detected cell states are due to batch effects. In the revised manuscript we address this question on two levels: 1) In Fig. S3A we show how the individual libraries and the library types (whole brain, dissected brain regions, FACSed whole brain) contribute to the UMAP. As expected, the FACSed samples show a strong enrichment of radial glia cells, but overall we observe a good mixing of the libraries in the radial glia cluster. While this data representation in principle contains the full information, it is hard to read because of the large number of samples. We therefore also performed a more targeted analysis: 2) In Fig. S3C (shown below as Fig. R1) we focus specifically on the radial glia cells in order to investigate the influence of batch effects on the cell states of interest. Furthermore, we limited the analysis to the seven non-FACSed whole brain libraries, for which the expectation (under ideal experimental conditions) would be equal representation of all cell states. We also note in the figure which version of the 10x Genomics Chromium kit was used for each library, since this would also be a potential source of batch effects. We find that all radial glia subtypes contain cells from multiple libraries. More specifically, the most abundant radial glia subtypes, as well as those of particular interest in this manuscript (snap25+, her4++, proliferating radial glia) were found in each sample (with the exception of sample b11, which overall contained hardly any radial glia). Overall, this analysis proves that the clustering of radial glia subtypes is reproducible and is not determined by batch effects.

Fig. R1 (included as Fig. S3C). Batch effect analysis of radial glia cells. Only whole brain, non-FACSed samples were included for which the expectation (under ideal experimental conditions) would be equal representation of all cell states. The versions of the 10x Genomics Chromium kit used for each library are listed, since this would be a potential source of batch effects. Top row: UMAP representation of radial glia, color represents sample ID and library type (left) or cluster (right). Bottom row: Bar graph representing samples contributing to each cluster. Each radial glia subtype contains cells from multiple libraries, suggesting that clustering of subtypes is not dominated by batch effects.

Fig.2

indication of replicates/pooled samples used are missing in the figure legend. This should be added. The RNAscope data should be quantified in addition to showing representative images.

Following the reviewer's request, we now performed a quantitative analysis of the RNAscope data for validation of the regional specificity of selected radial glia states (*apof* and *prss35* in mesencephalon and rhombencephalon; *AL954697.1* in mesencephalon and diencephalon; *crabp1b* in rhombencephalon and diencephalon). To this end, we now imaged all sections again under identical imaging conditions, which in our experience is necessary for reliable quantification of signal intensity. We now specify in the figure legend that we used 4 biological replicates (animals). For quantification we computed the fraction of the *gfap*:GFP positive area covered with RNAscope signal from the gene of interest (see Methods for details). This analysis, which is included in the revised manuscript in Fig. 2d and Fig. S4, and shown below in Fig. R2, confirmed the specific expression of *apof* in mesencephalon, of *prss35* in rhombencephalon, of *AL954697.1* in diencephalon, and of *crabp1b* in rhombencephalon.

Fig. R2 (included in Fig. 2d and Fig. S4). Quantification of RNAscope signal. The RNAscope signal for selected genes was quantified using Fiji software. The RNAscope signal was measured in the section area containing radial glia cells, as determined based on the *gfap:GFP* signal. The Region of Interest was applied to the thresholded RNAscope signal, and the fraction of the ROI covered by the RNAscope signal (Fraction of Area) was measured. In the plots, each dot represents one single animal (one representative section was analyzed per animal). The significance was tested using Mann-Whitney test and the p-value is indicated on the plot.

Fig.4

The concerns raised regarding RNA velocity: the authors write in their response that they are aware of the computational issues of RNA velocity and that they are working on improving them in the future but outside the scope of this manuscript. I can understand the argumentation of the authors however, have trouble seeing the value of a figure based solely on the trust on RNA velocity. Besides the concern regarding distinguishing the trajectories was not with regard to the UMAP representation, but rather to the trajectories shown, which are based on the authors interpretation and the line could as "clearly" also be drawn in different positions on the UMAP. Given that scSLAM-Seq is used to validate the results in Fig. 5 in this UMAP, the trajectories described in Fig.4 seem even less clear.

As discussed in the first revision, we agree with the reviewer that RNA velocity may be misleading due to several computational issues. However, it is equally important to note that RNA velocity has also been shown to work as expected in many biological settings, and has indeed proven to be a valuable tool for interpretation of cellular differentiation events. Furthermore, we wish to highlight that the scientific value of Fig. 4 is not solely based on RNA velocity analysis, but relies also on the profile of the UMAP representation in Fig. 4A, which shows a "donut" structure with an empty area in the center and two clearly separated transition zones between radial glia and neurons. We have rewritten the corresponding paragraph of the manuscript to highlight this observation, and we have changed the graphical representation of Fig 4A (shown below as Fig. R3) to increase clarity. Finally, we believe it is important to show conventional RNA velocity as well as "SLAM" velocity, in order to properly introduce our new "SLAM" velocity approach and to demonstrate that, in this case, both strategies lead to essentially the same results.

Fig. R3 (included in Fig. 4A). UMAP embedding of combined telencephalon dataset. Subsetted to radial glia and neurons, with velocity embedding streams derived from the stochastic model of splicing based velocities in scVelo. Subtypes of relevance for neurogenesis are highlighted among radial glia and neurons. Compared to the plot included in the previous version of the manuscript, we changed the colors of the cell clusters and decreased the density of the RNA velocity arrows, so the structure of the UMAP with two connecting regions between radial glia and neurons becomes more apparent.

We thank the reviewer for clarifying their question from the previous round of revision. We have modified our line of reasoning in the revised manuscript to highlight that our interpretation of two trajectories is based on three levels of analysis:

First, in the UMAP layout in Fig. 4A, we find that there are two separate areas connecting radial glia and neurons (the "donut" structure of the UMAP, with an empty zone in the middle, see Fig. R3). We would argue there is no major ambiguity in how these areas are separated. For the interpretation, the annotation of the different radial glia and neuronal states is crucial. These states are based on the clustering analysis of Fig. 1, and while the clustering depends on parameters chosen during data processing, we wish to highlight that there was no manual selection of transition points between different cell states in Fig. 4A. Of note, the line indicating the transition from radial glia to neurons was drawn manually, but is purely to guide the eye, and was not used in the analysis.

Second, the interpretation of these areas as transition zones or differentiation trajectories is supported by RNA velocity arrows (Fig. 4A, see also Fig. R3). While the arrows qualitatively appear to point from radial glia to neurons, the more important argument is the quantitative analysis of state transitions in Fig. 4B based on RNA velocity transition probabilities. This analysis suggests that radial glia can differentiate to neurons via two trajectories involving different radial glia states (proliferative versus non-proliferative differentiation). More specifically, the approach supports the interpretation that *her4++* radial glia can transition to proliferating radial glia, and further to newborn neurons, while *snap25a+* radial glia can transition directly to neurons but not to proliferating radial glia.

Third, and most importantly, in Fig. 4C we use CellRank to move beyond one-step transitions and infer full trajectories from radial glia to neurons. This approach combines identification of stable sets of cells in gene expression states with diffusion pseudotime and RNA velocity. This analysis confirmed possible differentiation paths from *her4++* radial glia to proliferating radial glia and to newborn neurons, and from *snap25a+* radial glia to neurons without proliferation (highlighted with arrows in Fig. 4C).

Of note, the two analyses in Fig. 4B and 4C also show additional transitions (besides *snap25+* radial glia to neurons and *her4++* radial glia to proliferating cells to neurons). This is to be expected for two reasons: First, the transitions are not directly measured but inferred based on transcriptomic similarity and RNA velocity, so a certain level of background signal is unavoidable. Second, and more importantly, we here analyze steady-state conditions in the unperturbed adult brain with only a limited amount of ongoing neurogenesis. It is therefore not surprising that, in addition to neurogenic differentiation, we also detect other transitions, including from global to regional radial glia states.

Fig.5

According to EV1 only 1 replicate of pooled brains was used per condition. This should clearly be stated in the figure legend

We have now included the number of replicates, as well as the number of cells per replicate, in the legend of Fig. 5D.

Reviewer #3:

I appreciate the work that the authors have put into revising the manuscript such as the addition of validation data using RNAscope. It is a nice piece of work and I support its publication. Some of my concerns still remain, but the authors might be right in their answer that this is beyond the scope of this work.

We thank the reviewer for the positive feedback and for the constructive comments in the first round of revision.

1st Feb 2024

Manuscript number: MSB-2023-11757RR

Title: Dissecting the spatiotemporal diversity of adult neural stem cells

Dear Philipp,

Thank you again for sending us your revised manuscript. We are now satisfied with the modifications made and I am pleased to inform you that your paper has been accepted for publication.

Best wishes,

Maria

Maria Polychronidou, PhD
Senior Editor
Molecular Systems Biology
